# Lignans of Sesame (*Sesamum indicum* L.): A Comprehensive Review

**DOI:** 10.3390/molecules26040883

**Published:** 2021-02-07

**Authors:** Mebeaselassie Andargie, Maria Vinas, Anna Rathgeb, Evelyn Möller, Petr Karlovsky

**Affiliations:** 1Molecular Phytopathology and Mycotoxin Research, University of Goettingen, Grisebachstrasse 6, 37073 Goettingen, Germany; annarathgeb@gmail.com (A.R.); evelyn.moeller@uni-goettingen.de (E.M.); 2Centro para Investigaciones en Granos y Semillas (CIGRAS), University of Costa Rica, 2060 San Jose, Costa Rica; maria.vinasmeneses@ucr.ac.cr

**Keywords:** sesame lignans, sesamin, sesamolin, sesamol, lignan biosynthesis, lignan glycosides, health-promoting properties, biological function, plant biotechnology

## Abstract

Major lignans of sesame sesamin and sesamolin are benzodioxol--substituted furofurans. Sesamol, sesaminol, its epimers, and episesamin are transformation products found in processed products. Synthetic routes to all lignans are known but only sesamol is synthesized industrially. Biosynthesis of furofuran lignans begins with the dimerization of coniferyl alcohol, followed by the formation of dioxoles, oxidation, and glycosylation. Most genes of the lignan pathway in sesame have been identified but the inheritance of lignan content is poorly understood. Health-promoting properties make lignans attractive components of functional food. Lignans enhance the efficiency of insecticides and possess antifeedant activity, but their biological function in plants remains hypothetical. In this work, extensive literature including historical texts is reviewed, controversial issues are critically examined, and errors perpetuated in literature are corrected. The following aspects are covered: chemical properties and transformations of lignans; analysis, purification, and total synthesis; occurrence in *Seseamum indicum* and related plants; biosynthesis and genetics; biological activities; health-promoting properties; and biological functions. Finally, the improvement of lignan content in sesame seeds by breeding and biotechnology and the potential of hairy roots for manufacturing lignans in vitro are outlined.

## 1. Introduction

Sesame (*Sesamum indicum* L.) is an ancient oilseed crop [1] cultivated in subtropical and tropical regions of Africa, Asia, and South America as a source of edible seeds and high-quality oil. The origin of cultivated sesame has not been conclusively identified [2]. Although Africa hosts most wild relatives of cultivated sesame, genetic arguments support the Indian origin of *Sesamum indicum* [3]. The Indian species *S. malabaricum* (syn. *S. mulayanum*) is the most likely progenitor ([3] and the references therein). To unite the crop and its progenitor under a common species name, Bedigian suggested new combinations *S. indicum* subsp. *malabaricum* for *S. malabaricum* and *S. indicum* subsp. *indicum* for the cultivated sesame [4]. In addition to *Sesamum indicum*, other species of *Sesamum* and a close relative *Ceratotheca sesamoides* are grown in Africa for seeds, and their leaves are locally used as vegetables [3]. Only *Sesamum indicum* (syn. *S. indicum* subsp. *indicum*), however, is regarded as a domesticated crop, and only seeds and oil of this species are traded internationally.

Major producers of sesame are Tanzania, India, Myanmar, China, Sudan, Ethiopia and Nigeria, in this order [5,6]. Landraces and locally grown varieties of sesame show conspicuous diversity, supposedly resulting from selection of variants by farmers and possibly also from repeated domestications [3]. In spite of its age and economical importance for local economies, sesame is regarded as an orphan crop and research devoted to sesame has been scarce; for instance, sesame is not mandated by any international crop research center [6].

Oil of *S. indicum* is valued for its sensory characteristics and resistance to rancidity [2,7,8]. Sesame oil also exerts antioxidative activity and possesses health-promoting properties, which are attributed to tocopherols, tocotrienols, and lignans [9,10,11]. Major lignans of sesame are sesamin and sesamolin (Figure 1). The total content of these two lignans in sesame seeds may exceed 1.4% (Table 1). Numerous minor lignans present in seeds in low concentrations and/or generated by chemical transformations during seed and oil processing have been described. Among them, sesamol, episesamin and samin (Figure 2) were studied extensively. Sesamol is a degradation product that is present in traces in unroasted seeds but occurs at high concentrations in roasted seeds and processed sesame oil (see Section 2.2). 

The economic value of lignans is reflected by patents covering purification, chemical transformations, and the use of lignans in health-promoting food additives and skin care components [12,13,14,15,16,17,18,19,20].

The concentration of lignans in seeds varies with the variety of sesame. High lignan content is a quality trait and an important target for sesame breeding. Inheritance of lignan content has only recently been systematically investigated (Section 4.3). 

Molecular markers for high lignan content are not available yet, though markers for the yield and many other agronomic traits have been established [21,22,23,24,25]. Progress in the genomics of sesame [26,27] and high-resolution genetic mapping [26,28,29,30,31,32,33] raise hopes that marker-assisted selection for lignan content as well as for a desirable ratio of individual lignans and their glycosides will be possible soon. 

Lignans are metabolites formed from two molecules of phenylpropanoids. In sesame, the synthesis of lignans involves the fusion of oxopropane side chains of cinnamyl alcohol into a furofuran core (3,7-dioxabicyclo[3.3.0.]octane) (Figure 3). These metabolites are designated furofuran lignans. Seven enzymes involved in the biosynthesis of lignans in sesame have been characterized (Section 3.1). Some of these enzymes catalyze multiple steps of the pathway. The genes encoding further enzymes, such as the dirigent protein involved in the initial step of the pathway and enzymes related to the formation of lignans that do not belong to the furofuran group (e.g., lariciresinol, secoisolariciresinol and matairesinol), were putatively identified in the genome of sesame based on sequence similarity.

Lignans of sesame attracted interest of nutritional scientists and health professionals because of their health-promoting activities (see Section 5.1 and Section 5.2) such as lowering blood glucose and cholesterol levels, prevention against cardiovascular diseases and cancer, and alleviation of postmenopausal syndrome [35]. The ability of sesamin to suppress tumor growth [36,37] suggests that sesamin might even be developed into a therapeutic agent. Sesame oil and lignans are components of creams and body oils [38,39]. Apart from the nutritional, cosmetic, and health-promoting use, sesame lignans and especially synthetically available sesamol served as potentiators of insecticides (see Section 5.4).

Several reviews of plant lignans are available [17,40,41,42]. Because of their nutritional importance, information about food lignans has been collected in several databases [43]. As these databases and reviews covered lignans from many plant species, space devoted to lignans of sesame was limited. In 2013, Dar and Arumugam [44] published a review entitled “Lignans of sesame: purification methods, biological activities and biosynthesis—a review” but they, too, covered lignans from many plants while the coverage of sesame was limited. A comprehensive review focusing on the lignans of sesame has been missing. This work attempts to review chemical, biological, and applied aspects of sesame lignans comprehensively. The literature was surveyed from the first reports on sesame lignans in the 1890th, including scarcely accessible historical texts. Representative examples were selected for finding documented in many reports. Controversial issues were critically examined, and several errors perpetuated in literature were corrected.

The Section 2 on lignan chemistry provides an overview of the chemical properties of lignans; their transformation and degradation during processing; the variation of lignan content in *S. indicum* and related species; and the purification, analysis, and total synthesis of lignans. In the Section 3 and Section 4 on biosynthesis and genetics, enzymes and genes of the lignan pathway are described, and published information on the inheritance of lignan content in sesame seeds is reviewed. In the Section 5, the biological activities and health-promoting properties of lignans are reviewed and their therapeutic potential is assessed. In the Section 6, biological functions of lignans in sesame plants are discussed. Finally, the potential of plant breeding and biotechnology for the improvement of lignan content in sesame seeds and for the production of lignans in vitro is assessed in the Section 7.

## 2. Chemistry of Sesame Lignans

### 2.1. Structures and Chemical Properties of Lignans of Sesamum indicum

#### 2.1.1. Aglycons

Haworth introduced the term lignan in 1936 for phenolic metabolites of plants that consist of two n-propylbenzene moieties [45]. Phenylpropanoid monomers are connected via β-atoms of the propane chains [34]. According to the cyclization pattern and the presence and location of oxygens, lignans are divided into eight classes [34]. Major lignans of sesame (Figure 1) belong to the furofuran family. In furofuran lignans, the oxopropane side chains of phenylpropanoid building units are fused into 3,7-dioxabicyclo[3.3.0]octane (Figure 3). Minor lignans of *Sesamum indicum* (Figure 2) belong to the furofuran, tetrahydrofuran, and butyrolactone classes; further lignans are produced by the degradation and transformation of furofuran lignans. Chemical properties of lignans of *S. indicum* and their content in seeds of sesame are shown in Table 1.

All *Sesamum* species studied so far produce lignans. Sesamin and sesamolin were detected in most species and we assume that all *Sesamum* species produce these two lignans, though most species accumulate them in lower levels than *S. indicum*. In addition to sesamin and sesamolin, some sesame species produce unique lignans not occurring in *S. indicum* (Table 2). These species are of commercial interest as a source of enzymes and genes for the engineering of lignan biosynthesis (K. Dossa, personal communication).

The major lignans of sesame—sesamin and sesamolin—are also the oldest lignans described. Sesamin was isolated by James Fowler Tocher in 1890th in Aberdeen, Scotland, from acetic acid extract of sesame oil [46,47,48]. The structure of sesamin was elucidated in 1939 at the University of Würzburg, Germany [49] and its absolute configuration was determined in 1960 in Heidelberg, Germany [51].

The second major lignan of sesame is sesamolin [50,72]. The name sesamolin was coined in 1928 by W. Adriani [69] for a crystalline compound with a melting point of 94 °C, which was isolated from sesame oil for the first time in 1903 in Italy [68]. The structure of sesamolin was determined in 1955 at the University of Sheffield [70].

Further lignans often reported from sesame, that are are present in smaller amounts, are pinoresinol [76], sesaminol [73,85], sesamolinol [75], episesaminone [86], matairesinol [77], and episesamin [73].

Mixtures of lignans containing sesamolactol were extracted from several plants other than sesame since the 1980th but the structure of (-)-sesamolactol and its presence in the perisperm of sesame seeds were only established in 2006 [96]. Lariciresinol was originally purified from resin of spruce [97], but in 2005 it was also found in sesame seeds [77]. Sesamol, sesaminol, sesaminol epimers, and episesamin are degradation products, which are found only in traces in unroasted seeds and oil (see Section 2.2). Sesaminol was also reported in unroasted fully matured and germinating seeds [85].

#### 2.1.2. Lignan Glucosides

Sesaminol and pinoresinol occur in sesame seeds mainly in glycosylated forms (Table 3 and Figure 4). Sesame oil contains aglycons and some monoglycosylated lignans, while most of di- and tri-glycosylated lignans remain in oil-free meal after oil extraction. Major glycosylated lignans in sesame seeds are di- and triglucosides of pinoresinol [76,98], mono-, di- and triglucosides of sesaminol [99], and sesamolinol diglucoside [100]. The structures of glucosylated lignans of sesame are shown in Figure 4. Enzymatic deglucosylation of lignan glucosides is reviewed in Section 2.2.2.

#### 2.1.3. Does Sesame Contain Pinoresinol Monoglucoside?

The presence of pinoresinol monoglucoside in sesame is the first out of two controversial questions regarding pinoresinol glucosides in sesame. Some authors claimed that sesame seeds contained pinoresinol monoglucoside, but we were unable to find published data substantiating their claim. Moazzami et al. [100] listed pinoresinol mono-, di- and triglucosides as components of sesame seeds in the introduction section of their paper, citing Katsuzaki et al. [76], but the cited paper does not support the presence of pinoresinol monoglucoside in sesame. The authors repeated the claim in another paper published in the same year [101], citing the same work by Katsuzaki et al. [76] and two papers of their own [100,102]. None of the cited papers supported the claim.

Pathak et al. [103] listed pinoresinol monoglucoside among the component of sesame seeds in their review of bioactive compounds in sesame, citing the seminal work by Katsuzaki et al. [52] and the publication by Moazzami et al. [102] mentioned above. None of these publications dealt with pinoresinol monoglucoside. Dar and Arumugan [44] listed pinoresinol monoglucoside among the lignans of sesame, citing several publications, yet none of the cited papers supported the claim. Gerstenmeyer and coworkers [104] reported that they presumably detected pinoresinol monoglucoside in sesame seed by HPLC-MS based on the *m*/*z* value of its molecular ion, yet they could not verify the identity of the analyte due to the lack of a standard.

In summary, the critical review of literature revealed that the presence of pinoresinol monoglucoside in sesame is not supported by data. We hypothesize that a sensitive analytical method will identify traces of pinoresinol monoglucoside in sesame seeds as intermediate of the synthesis of di- and triglucosides or as a product of partial hydrolysis. Pinoresinol monoglucoside is produced in plants other than sesame such as *Forsythia* [105], and prune [106].

#### 2.1.4. Bias of Biomedical Research on Pinoresinol Diglucoside

In their seminal work, Katsuzaki and coworkers [52] characterized three diglucosides of pinoresinol: two diglucosides with both glucose molecules attached to the same methoxyphenol moiety and another diglucoside with glucose molecules attached to different methoxyphenols. These three diglucosides were present in sesame seeds in comparable concentrations. However, only the latter compound, with glucose molecules attached to different methoxyphenol moieties of pinoresinol, was used in all biomedical studies on pinoresinol diglucoside that we are aware of (e.g., [107,108,109]).

Pinoresinol di-*O*-β-d-glucoside was used as the only standard in many analytical methods for the quantification of pinoresinol diglucosides (e.g., [110]). Furthermore, protocols for the quantification of total pinoresinol relying on enzymatic deglucosylation that were validated only with pinoresinol di-*O*-β-d-glucoside (e.g., [111]) might fail with other pinoresinol diglucoside isomers.

The use of a single isomer as a standard for the quantification of pinoresinol diglucosides likely lead to an underestimation of pinoresinol diglucoside content in sesame [112,113] and other species [114]. Pinoresinol diglucosides other than di-*O*-β-d-glucoside were rarely analyzed [102].

What is the reason for this bias? To our knowledge, only pinoresinol di-*O*-β-d-glucoside is available commercially, e.g., from Sigma-Aldrich, Selleckchem, Cayman Chemicals, ChemFaces, Adooq Bioscience, Apexbio, and other suppliers. The companies that specified the source always named *Eucommia ulmoides*. We speculate that all commercial pinoresinol diglucoside originates from the bark of *E. ulmoides* rather than sesame [115]. *E. ulmoides* is a tree cultivated in China for the production of gutta-percha [116]. Extracts of the bark are used in Chinese traditional medicine (杜仲 [dù zhòng], in English known as tu-chung) [117]. While all isomers of sesaminol diglucosides are available from Japanese companies Nakalai Tesque and Nagara Science, we are not aware of a commercial source of pinoresinol 4′-*O*-β-d-glucopyranosyl-β-d-glucopyranosides. We assume that the availability of pinoresinol di-*O*-β-d-glucoside from *E. ulmoides* on the market accounts for the bias of research on biomedical effects of pinoresinol diglucoside. Comparative studies with all three diglucosides are highly desirable.

### 2.2. Transformation and Degradation of Sesame Lignans

#### 2.2.1. Transformation and Degradation of Lignans during Seed and Oil Processing

Seeds of sesame are often roasted to improve their sensory properties and sesame oil is subjected to industrial raffination, which includes alkaline saponification and bleaching with acidic clay. Several chemical transformations of lignans occur during these processes (Figure 5).

The most important transformation is the production of sesamol from sesamolin. The term sesamol was coined by Hans Kreis in 1903 for yet unidentified phenolic product of sesame responsible for color tests used to identify sesame oil [118] (see also Section 2.5.2). Kreis was apparently unaware of Villavecchia and Fabris work from 1893 [119]. The structure of sesamol was established four years later [79,80]. In unroasted seeds and raw oil, sesamol is present in traces or undetectable [73,120,121]. Conversion of sesamolin to sesamol is catalyzed by acidic clays used for decoloration (bleaching) of sesame oil and by heating [87,122,123,124,125].

Yoshida and Tagaki [126] investigated the effect of temperature on the conversion of sesamolin to sesamol in sesame seeds. Essentially all sesamolin was converted to sesamol after 25 min at 250 °C (Figure 6). At high temperatures, sesamol dimerizes into sesamol dimer (Figure 5), which possesses antioxidative activity [127], but the concentration of sesamol dimer in refined oil is very low [123]. In the presence of FeCl_3_, oxidation of sesamol by oxygen lead to a mixture of complex conjugated dimers [128], which exerted cytotoxic activities [128,129]. However, the reaction conditions used (10 days at 40 °C in the presence of 10 mol % FeCl_3_) do not occur during processing and storage of sesame oil. Realistic conditions for industrial-scale epimerization of sesamin, an apparatus, and a procedure for preparative enrichment of episesamin were protected by a patent [130].

Sesamolin is transformed into sesaminol and its epimers during bleaching of sesame oil [73,99] (Figure 5A). Bleaching also leads to the epimerization of sesamin (Figure 5B) [73,123]. The mechanism of these transformations has not been conclusively established. Based on their study on the conversion of sesamolin to sesaminol under anhydrous conditions catalyzed by sulfonic acid and on Fukuda’s original work [131], Huang and coworkers [132] suggested a two-step mechanism. According to their hypothesis, protonated sesamolin brakes down into sesamol and oxonium ion, both of which subsequently recombine into the products. Depending on the orientation of molecules engaged in the reaction, sesaminol, its epimers, or sesamolin epimer is produced [132]. We suggest that in sesame oil highly reactive oxonium ions would react with major components of the matrix, which are present in large excess, before re-joining with sesamol. We assume that an intramolecular rearrangement of protonated sesamolin might account for the transformation; in any case, the mechanism has to be elucidated experimentally.

Sesamolin by itself has no antioxidative activity yet its transformation products sesaminol and sesamol are strong antioxidants, believed to be responsible for a large part of antioxidative effects of sesame oil. Heating as part of industrial processing was therefore investigated with the aim of increasing the content of antioxidative lignans [126].

#### 2.2.2. Enzymatic and Alkaline Hydrolysis of Lignan Glucosides

Deglucosylation of lignan glucosides by intestinal bacteria [133,134] is the first step of lignan metabolism in the digestion track of mammals (Section 2.2.3). In the laboratory, deglucosylation is used to determine total aglycon concentration (Section 2.5.2) and to maximize the yield of aglycons in preparative purification (Section 2.5.3).

Deyama [115] reported successful hydrolysis of pinoresinol diglucoside to aglycon by a commercial β-glucosidase. Mono- and diglucoside of sesamolinol cannot be hydrolyzed by commercial β-glucosidases, probably because steric hindrance prevented enzymatic catalysis [98]. Glucosides of lignans other than sesamolinol were successfully hydrolyzed by glycosidases [52,76]. Deglycosylation of lignan glucosides was also achieved with a mixture of glycosidases and cellulases [54,135]. Park and coworkers reported that the main hydrolysis product of sesaminol triglucoside in their hands was monoglucoside rather than aglycone even after prolonged treatment with a mixture of β-glucosidase and cellulase [136]. Peng and coworkers, in spite of thorough optimization of the procedure with the same combination of enzymes, achieved only a 50% yield of pure aglycone [137]. Different properties of the enzymes used in different labs may explain the contradiction; unfortunately, the sources of the enzyme have often not been reported. Gerstenmeyer and coworkers [104] reported that prolonged incubation with β-glucuronidase/arylsulphatase from *Helix pomatia* (Roche, Mannheim, Germany) and cellulase Onozuka R-10 (Merck, Darmstadt, Germany) completely hydrolysed all lignan glucosides, while treatment with cellulase alone was not sufficient.

To overcome the limitation of commercially available glycosylases, researchers from Kiyomoto Co. in collaboration with two universities in Japan designed a smart screening strategy to identify microbial sources of a glucosidases suitable for the hydrolysis of lignan glycosides, especially sesaminol triglucoside [138]. They collected samples of decaying sesame oil cake, extracted them with chloroform, and analyzed the extract for traces of sesaminol. One of the positive samples yielded a strain of *Paenibacillus* sp. that hydrolyzed sesaminol triglucoside. The glucosidase responsible for the reaction was purified, partial amino acid sequence of the protein was determined, and the gene was cloned and expressed in *E. coli*. The recombinant enzyme hydrolyzed sesaminol triglucoside to pure aglycone. The authors applied for patent protection [139]. Later on, they identified a second glucosylase produced by the same bacterium, which was specific for the β-1,2-glucosidic bond of sesaminol triglucoside, producing sesaminol diglucoside [140]. Gaya and coworkers [141] investigated β-glucosidase from *Lactobacillus mucosae*, which deglucosylated secoisolariciresinol diglucoside into secoisolariciresinol, and successfully expressed the gene in food-grade lactic bacteria for the deglycosylation of lignans in food.

Instead of treatment with glycosidases, alkaline hydrolysis can be used to convert lignan glucosides to aglycons. Various conditions for alkaline hydrolysis have been reported in literature, such as refluxing with 1 M potassium hydroxide in ethanol [142], treatment with 9 M sodium hydroxide in water at room temperature overnight [143], and treatment with sodium methoxide in pure methanol (3 h at 40 °C with sonication) [104]. A combination of alkaline hydrolysis (0.3 M NaOH in 70% methanol, incubated for 1 h at 60 °C) with a subsequent enzymatic hydrolysis using β-glucuronidase/sulfatase from *Helix pomatia* has also been used [144]. Secoisolariciresinol diglucoside was hydrolyzed successfully by incubation with 1 M sodium or potassium hydroxide for 4 to 24 h at room temperature [145].

#### 2.2.3. Transformation of Ligans in the Body of Mammals

In the digestive track of animals, intestinal bacteria remove glucose from lignan glucosides [133,134] and transform lignan aglycons into metabolites designated enterolignans [40,146]. Major enterolignans produced by microorganisms in the human digestive track are enterodiol and enterolactone [147,148,149,150,151]. The conversion of lignans to enterolignans was also demonstrated in vitro in cultures inoculated with human fecal inoculum [152] and in axenic cultures of bacteria isolated from human intestine (e.g., [151,153]). The conversion involves four steps: deglycosylation, demethylation, dehydrogenation, and dehydroxylation. In addition, one or two reduction steps are involved, depending on the type of lignan [108,151]. Because of the potential of phytoestrogens to ameliorate menopausal syndrome (Section 5.2.2), the conversion of food lignans into enteroligans by female intestinal microflora attracted research interests. Corona and coworkers [144] compared the conversion of secoisolariciresinol, lariciresinol, pinoresinol, and matairesinol from an oilseed mixture by fecal microflora of young and premenopausal women. They found that the fecal microflora of young women generated mostly enterolactone while the microflora of premenopausal women generated mostly enterodiol. The results were reported with a precision of up to six significant figures, yet the relative standard deviations of most values exceeded 100%, suggesting that the results should not be overrated.

Different steps of lignan transformation in the human gut are catalyzed by different bacterial species including *Clostridium* spp., *Bacterioides* spp., *Eubacterium* spp. [150]. The final dehydrogenation of enterodiol into enterolactone was catalyzed by a new strictly anaerobic bacterium [148], which was later characterized and named *Lactonifactor longoviformis* [154]. *Ruminococcus* sp. isolated from human intestine also transformed enterodiol into enterolactone [155].

Cytochrome P450 oxidases in the liver of mammals transform lignans by opening and demethylating methylenedioxy-moieties, converting them to vicinal dihydroxyphenol (catechol) derivatives. These reactions were studied in liver homogenates of rats [156], in human liver microsomes, and with human liver enzymes expressed in yeast [157]. Sesamin monocatechol is eventually glucuronated and methylated [158], and the metabolites are excreted in the bile and urine [159].

### 2.3. Total Synthesis and Industrial Production

Total synthesis of lignans will be briefly outlined in this section. Industrial synthesis of sesamol, introduced in the 1950th [12], provides sesamol at a much lower price than the extraction from sesame oil. Several synthetic strategies were exploited, continuously improving the industrial production [13,160]. A number of synthetic routes for native furofuran lignans have been reported since the 1990th, but none proved suitable for industrial use. The main challenge for the chemical synthesis of lignans was the control of stereochemistry of the furofuran core. The synthesis of tetrahydrofuran lignans is less complex; lariciresinol [82] was synthesized in 1994. Lignans of both tetrahydrofuran and furofuran families including pinoresinol, piperitol, and sesamin were obtained by radical cyclization of epoxides [78]. In the last decade, furofuran lignans have been the target of new synthetic efforts. Electrochemical asymmetric oxidative dimerization of cinnamic acid was used in a biomimetic approach to synthesize sesamin [161]. An elegant bioinspired yet not biomimetic approach to furofuran lignans was the exocyclization of biaryl cyclobutane, which afforded pinoresinol [162]. Sesamin and sesaminol were also synthesized by crossed aldol reaction with a quinomethide intermediate [163]. Asymmetric synthesis of furofuran skeleton [164], the total synthesis of tetrahydrofuran lignans [165], and general synthetic approaches to furofuran lignans were recently reviewed [166].

Patent protection for the total synthesis of (+)-sesamin and other furofuran lignans based on the alkylation of chiral epoxides have recently be sought [167], indicating that the chemical synthesis of lignans matures towards industrial use.

### 2.4. Stabilization of Fats by Sesame Oil and Unexpected Discovery of Sesame Lignans in Diverse Oils

Antioxidative activity of lignans and tocopherol protects fats from spoilage by rancidification. Therefore, small amounts of sesame oil used to be added to animal fats, vegetative oils, and shortenings to stabilize them [168,169,170]. The method was protected by numerous patents [171,172,173,174].

A recent revelation of the presence of sesame lignans in diverse vegetable oils puts a new spin on the topic [175]. Caraway, rapeseed, hemp, peanut, sunflower, pumpkin, poppy, and other edible oils were found to contain sesamin and sesamolin at the same ratio that is known from sesame. The oils were not declared to contain any sesame oil. The authors of the study suggested that sesame lignans in these oils resulted from unintended contamination; pointed out the risk for consumers allergic to sesame; and recommended cleaning the processing equipment thoroughly [175]. We would not rule out that certain companies may add undeclared sesame oil to their products intentionally to extend their shelf life, which would violate food law and possibly also infringe patent rights. Therefore, including sesame lignans in food safety monitoring appears advisable even for oils and fats that are not declared to contain sesame oil. Large number of suspicious products can be pre-tested using a simple colorimetric assay (e.g., [176], see also Section 2.5.2). This may be particularly useful in environments where adulteration of fats is a common practice [177].

### 2.5. Extraction, Analysis, and Purification

#### 2.5.1. Extraction of Lignans from Seeds and Oil of Sesame

The polarity of lignan molecules is low to medium, but extraction protocols have to take into account that most lignans are glycosylated and the solubility of di- and tri-glycosides in organic solvents is limited. The extraction of sesame seeds with 80% ethanol, which is suitable for aglycons as well as glycosylated lignans, was developed in 1998 [53]. In this work, Ryu et al., however, defatted crashed seeds with n-hexane before extraction [53]. Defatting seeds before extraction has sporadically been used until recently (e.g., [178]). The levels on unglycosylated lignans reported in these studies were likely underestimated because lignans dissolve in n-hexane to a certain extent. Many groups actually extracted lignans from sesame into n-hexane [54,87,179,180], or used lignan solutions in n-hexane for crystallization [181]. The solubility of lignans in pure n-hexane is limited, as shown by [142], who reported precipitation of sesamin and episesalatin from n-hexane. Therefore, refluxing in Soxhlet apparatus [55,142,180,182] or repeated extractions with n-hexane [56,183] were typically used. n-hexane is certainly not an ideal solvent for lignans. Defatting seeds with n-hexane (or cyclohexane, which has similar properties and is preferable because of lower toxicity) can be safely used before extraction of glycosylated lignans [52,57,100,102].

Several solvents have been used for the extraction of lignans for analytical purposes. Pure methanol [88], mixtures of methanol and chloroform [121,126] (which do not extract glycosides), a mixture of ethanol with water [58], and a mixture of ethanol with acetate buffer [104] have been used. Other extraction solvents used in the past included heptane or hexane with isopropanol in a 1:3 ratio [54,101], and acetone/water [147,184]. Extraction with 80% ethanol without previous defatting [58] is the standard protocol today (e.g., [59,60,61]). 80% methanol is also occasionally used [94,185]. Apart from aglycons and monoglucosides, 80% ethanol (and likely 80% methanol) extracts di-glucosides. Tri-glucosides were successfully extracted into 80% ethanol by some authors [52,76,100] while other found tri-glucosides in the insoluble residues after extracting seeds with 80% ethanol [53]. 70% acetone is an interesting alternative to 80% ethanol because it is suitable for the extraction of all lignans [186] as well as their conjugates [147].

#### 2.5.2. Color Tests and Chromatographic Methods for Lignans Analysis

Before chromatography became established in the analysis of sesame oil, color reactions and photometry in UV light were used to detect certain lignans and semi-quantitatively estimate their concentration. Villavecchia’s colorimetric test [119] was established to distinguish sesame oil from other oils and to differentiate butter from margarine (Figure 7), which was labeled with 5–10% sesame oil in the first half of the 19th century [122,187]. Villavecchia test became the official method of the American Oil Chemists’ Society for the detection of sesame oil in vegetable and animal oils and fats [188]. Because Villavecchia test responds to sesamol, it is suitable for the quantification of this lignan [189,190]. When Budowski and coworkers [81] recognized that sesamolin produced the same color reaction because it was converted into sesamol under acidic conditions of the test, they developed a photometric test based on light absorption of the product of furfural-sulfuric acid reaction with sesamol at 518 nm. They also established an analytical method for sesamin based on UV absorption of oil after removal of sesamol by treatment with alkali [50]. Suarez et al. [62] combined their method with the Villavecchia reaction for the determination of the content of sesamol, sesamolin and sesamin in oil. A new method for rapid estimation of total lignans in sesame oil relying merely on the light absorption at 288 nm was developed as recently as in 2015 [180].

After chromatography became widely available as an analytical method, colorimetric methods became obsolete [191]. In spite of that, the Villavecchia test was used until recently for the estimation of the content of sesame oil in pharmaceuticals because it allowed for a high throughput. Even the Budowski/Suarez test was occasionally used for the determination of sesamol in oil until recently [183]. A new colorimetric test was developed in 2005 for the detection of adulteration of edible fats with sesame oil [176]. The test relies on the reaction of thiophene carboxaldehyde with sesamol under acidic contions, and it can detect 0.1% sesame oil in other oils or fats.

Normal-phase chromatography on analytical columns for the analysis of sesamin, sesamol and sesamolin was available since the 1950’s [63]. Thin-layer chromatography (TLC) and especially two-dimensional TLC [192] and HPTLC [193] were used in parallel with normal-phase liquid chromatography [123] untill recently. Gas chromatography coupled with mass spectrometric detection (GC-MS) was introduced into lignan analysis in the 1990s [76,142] and is still used [194]. A thorough comparison of TLC, GC-MS, and HPLC-UV was carried out by Kamal-Eldin and coworkers [142]. Normal-phase HPLC is still used occasionally, as recently shown for sesamin, sesamol and sesamolin [195], but reverse-phase columns eluted with water-methanol or water-acetonitrile gradients have superseded normal-phase chromatography in the meantime in most methods for lignan analysis [101,142,194,196]. Fluorescence and UV light absorption as detection signals [53] are being gradually replaced by tandem mass spectrometry (e.g., [104,196,197]). A comprehensive HPLC-MS/MS method covering all major lignans of sesame has not been developed yet. Electrochemical methods for sesame lignans have recently been established. For instance, voltametric methods for the determination of sesamol directly in sesame oil [198] and in acidic solutions [199] were described. Whether these methods can compete with HPLC-MS/MS in routine analysis remains to be seen.

#### 2.5.3. Purification of Sesame Lignans

All solvents used for the extraction of lignan for analytical purposes are suitable for preparative purification, but other solvents have also been used, especially in industrial production. 80% ethanol was used for aglycons as well as glycosides of lignans [52,76]. Acetone [71,200], hot pure methanol [57,201,202,203], and 80% methanol [57] were also used. Standard methods of natural product chemistry used for the purification of lignans included differential crystallization [64,200,204], column chromatography [52,53,182,205], counter-current chromatography [83,178,196], and preparative TLC [64,206]. In spite of the limited suitability of n-hexane for the extraction of lignans (see Section 2.5.1), hexane was occasionally used for preparative purification of lignans (e.g., [207]). The limited solubility of the target compounds in hexane can be compensated by a large solvent-to-sample ratio and protracted extraction in Soxhlet apparatus, which continuously exposes the sample to pure solvent generated from condensing vapors.

Liquid-liquid extraction can be used to enrich particular lignans in extracts before purification. For instance, partition between n-hexane and water enriched piperitol and pinoresinol in the water phase, and subsequent partition between water and ethyl acetate enriched both compounds in the ethyl acetate phase [202]. Liquid-liquid extraction into ethyl acetate was used for the purification of aglycons and mono- and diglucosides but triglucosides cannot be extracted efficiently into ethyl acetate (see Section 2.5.1).

Some researchers added butylated hydroxytoluene to extraction solvents to prevent oxidation [121,126,179] but most labs did not regard this as necessary. For instance, Williamson and coworkers [208] have not added butylated hydroxytoluene to extracts of sesame seeds for the purification of lignans, though they added butylated hydroxytoluene to the same extract for the purification of other oxidation-sensitive metabolites.

An interesting liquid-liquid extraction of lignans from sesame oil into unconventional solvent γ-butyrolactone was developed for the production of pyrethrin synergists by the Norda Essential Oil and Chemical Company in New York [209]. According to this method, a mixture of sesame oil and γ-butyrolactone is heated to 130 °C until a homogeneous solution is obtained. After cooling to 60 °C, solvent and oil separate. Lignans are obtained from the solvent layer after removal of γ-butyrolactone by distillation.

Supercritical extraction of sesame lignans with carbon dioxide liquefied by a high pressure (150 to 350 bar) [210] was developed for industrial production of lignans. The extraction is carried out at ambient temperature under nearly anaerobic conditions. Extraction of lignans with supercritical butane is protected by a Chinese patent [211].

Seed or oil extracts for the purification of lignans can be treated with glucosidases or exposed to alkaline conditions to hydrolyze lignan glucans, increasing hereby the yield of aglycons (see Section 2.2.2).

### 2.6. Variation in Lignan Content among Accessions of Sesame

The content of lignans in sesame seeds varies by an order of magnitude. Differences in lignan content among varieties and accessions have been documented in dozens of studies (e.g., [54,65,73,101,152]), but a comprehensive overview of these results is missing.

All furofuran lignans in sesame originate from the same pathway, therefore their concentrations are supposed to correlate. Within a set of 65 varieties developed by the sesame breeding company Sesaco Corporation, which included varieties with white, yellow, brown and black seeds, strong positive correlations were established between the content of sesamin and sesamolin (R^2^ = 0.69), as well as sesaminol and sesamolinol (R^2^ = 0.53) [101]. Interestingly, correlations between sesamin and sesaminol, sesamin and sesamolinol, sesamolin and sesaminol, and sesamolin and sesamolinol were negative (R^2^ = 0.37, 0.36, 0.35, and 0.46, respectively) [101]. Our results obtained with a set of 25 sesame varieties from different parts of the world have not confirmed negative correlations of sesamin with sesaminol nor sesamolin with sesaminol (unpublished data). The comparison of sesamin and sesamolin content in 21 hybrids of Thailand varieties of sesame revealed a strong positive correlation [185]. Similar results were reported from Japan [212], China [60], and India [61,94,213]. Tashiro and coworkers [214] reported very tight positive correlations between sesamin and sesamolin content in 42 varieties from Asia and Latin America, covering a wide range of agronomic characteristics. The content of sesamin was significantly higher in white and brown seeds than in black seeds; no consistent trend was found for sesamolin. The correlation between the content of sesamin and sesamolin in black and brown seeds was tight (r = 0.778 and 0.903, respectively); weaker correlation between sesamin and sesamolin was found in white seeds (r = 0.386).

Comparison of lignan content in 43 varieties of sesame from all climatic zones in India showed that the content of sesamin and sesamolin was higher in black seeds than in white and brown ones [61]. Another study from India found no relationship between seed color and lignan content [94]. In Chinese varieties of sesame, lignan content was higher in white seeds than in black ones [60]. Thus, it appears that seed color and lignan content in sesame are unrelated; the associations reported in some studies may be due to limited genetic diversity in the germplasm collections used.

Figure 8 shows the content of sesamin and sesamolin as reported for varieties and accessions from three continents. The source data and references for this overview are provided in Appendix A. The overview confirms a positive correlation between the content of sesamin and sesamolin. No relationship between the geographical origin and lignan content is apparent, which was expected because sesame is an old crop with a long history of seeds trade [1,215].

The relationship between the genetic relatedness of sesame accessions and their lignan production has not been investigated. It is only known that patterns of metabolic diversity in general are incongruent with the genetic relatedness in sesame [216]. This situation is common in plants [217,218,219,220] and we assume that it holds for lignans of sesame, too. Secondary metabolite production is subjected to a strong selection pressure while surveys of genetic diversity rely on neutral markers such as amplified fragment length polymorphism AFLP [221] and microsatellites [222]. Patterns of genetic diversity obtained with these methods cannot be used as a predictor of lignan production. Molecular markers linked to the *loci* affecting lignan synthesis are needed; the development of such markers is described in Section 4.3.

Comparison of varieties regarding their lignan content in tissues other than seeds has only been reported in a single study. According to Kareem and coworkers [223], the differences in sesamin content of roots and hairy root among accessions and varieties of *Sesamum indicum* were larger than the differences among the sesamin content in seeds (Table 4). Lignans other than sesamin were not detected in roots or hairy roots [223].

### 2.7. Lignans in Other Tissues and Organs of Sesamum indicum

Virtually all research on the lignans of sesame was carried out on seeds and oil obtained from seeds. The synthesis of lignans is, however, not limited to seeds (Table 4). Callus cultures from *Sesamum indicum* accumulated sesamin [224,225,226] and sesamolin [225,226]. According to Ogasawara and coworkers [226], sesamin and sesamolin were reported from callus cultures for the first time in 1987 by the Biotechnology Research Laboratory of Kobe Steel Ltd., Tsukuba, Japan [227,228,229] (in Japanese). Sesamol, sesaminol and sesamolinol were not detectable in callus [225,226]. The lack of sesamol in callus cultures was confirmed in an independent study [230]. Hairy root cultures produced sesamin but no other lignans [231]. Interestingly, the content of sesamin in hairy roots was higher than in roots [231].

Sesamin was also reported from sesame leaves [232] and roots [231]. Sesamin concentrations in leaves were ca. 5000-times lower than the concentrations in seeds of the same varieties [232]. In the roots of most of 25 sesame varieties studied, the concentration of sesamin has not exceeded 10 mg/kg, which is 70 to 900-times less than the sesamin content in seeds (cf. Table 1 and Figure 8). Kareem [231] did not find lignans other than sesamin in the roots. Fuji and coworkers [233], studying non-lignan metabolites of sesame, reported that they found sesamin and sesamolin in leaves, stem, root, and flower of sesame. The analytical method used was HPLC with both UV absorption and ESI-MS detection. The authors claimed that they validated their results using HPLC-ESI-MS/MS. The publication, however, does not show any analytical data on lignans [233]. We have not found sesamin or sesamolin in any part of sesame plants except for capsules with seeds, leaves, and roots (unpublished data).

As shown in Table 4, lignans do not accumulate in organs and tissues other than seeds to appreciable levels. The repertoire of lignans in callus tissue is limited to sesamin and sesamolin, while sesamin also accumulated in the roots and hairy roots of sesame [231]. Sesamin content in the leaves and young and old callus amounted to less than 5% of the levels in seeds; similarly, sesamolin content in young and old callus amounted to less than 5% of the levels reported in seeds (Table 4) [225,226,232]. Lignans other than sesamin and sesamolin were not reported from organs other than seeds nor from tissue cultures of sesame.

### 2.8. Lignans in Wild Relatives of Sesame

Several lignans known from *S. indicum* and new lignans not known from cultivated sesame have been found in wild relatives of sesame (Table 2). In 1951, Pearman and coworkers [234] as cited by [66] reported sesamin from *Sesamum angolense*. Bedigian and coworkers [66] screened several wild species of sesame for the presence of sesamin and sesamolin. They confirmed the presence of sesamin and sesamolin in *S. angolense* and found these two lignans also in the seeds of *S. angustifolium*, *S. calycinum*, and in *S. orientale var. malabaricum*. Only sesamin was found in *S. latifolium* and *S. radiatum*. In addition, trace amounts of sesamin and sesamolin were also observed in *S. petaloides* and *S. capense,* respectively. Bedigian and coworkers [66] also reported the presence of lignans in the seeds of related genera: sesamin was found in *Ceratotheca sesamoides*, and both sesamin and sesamolin were found in *Sesamothamnus busseanus*. In addition, the authors found trace amounts of sesamolin in *Ceratotheca sesamoides* and trace amounts of sesamin in *Ceratotheca triloba*, *Holubia saccata, Pedalium murex,* and *Pretrea zanguebaricum*.

Lignans from wild relatives of sesame that are not present in *S. indicum* are (+)-sesangolin, (+)-alatumin, (+)-2-episesalatin, and (+)-7′-episesantalin (Table 2). Furofuran lignan (+)-sesangolin was purified from *S. angolense* in a search for new synergists because seed oil of *S. angolense* possessed unusually high synergistic activity with pyrethrum [91]. Sesangolin was later reported from *S. angustifolium* [87], *S. alatum* [88], and *S. radiaum* [92]. Both alatumin [89] and episesalatin [90] were found in *S. alatum*. The most recent contribution to the diversity of sesame lignans is (+)-7′-episesantalin, which was purified from *S. radiatum* [92].

Apart from a limited specificity of TLC used for lignan analysis in old studies, some studies of lignans in wild relatives of cultivated sesame were plagued by erroneous taxonomic assignments. *S. latifolium* was mistaken for *S. radiatum* in two studies [87,179] published in 1994, as the author noted in her subsequent publication from 2010 [95] (see footnote 4 in Table 2). Thus, papers published between 1994 and 2010 might have spread the wrong data; an example is an industrial compendium on oils [235]. Recently, a Suntory Foundation for Life Sciences and their collaborators in Japan [92] re-investigated the content of lignans in seeds of *S. radiatum*, confirmed sesamin and sesamolin, and discovered the sesangolin isomer (+)-7′-episesantalin. Likewise, they re-discovered the presence of (+)-sesangolin, apparently not aware of the publication by Su Rho Ruy and coworkers from 1993 in Korean language, in which sesangolin was reported from *S. radiatum* for the first time [88].

Some controversies about taxonomic assignments in these studies remain unresolved. For instance, in 2010 Kamal-Eldin [95] reported the content of sesamin and sesamolin in *S. latifolium* as published in 1988 in a Japanese journal [72], but in the cited publication *S. radiatum* rather than *S. latifolium* was named. Kamal-Eldin [95] did not explain why she assumed that *S. radiatum* in this publication [72] was actually *S. latifolium*.

Future studies relying on HPLC coupled with tandem mass spectrometric or high-resolution mass-spectrometric detection will likely render old results based on TLC obsolete. Apart from the accessibility of more powerful analytical techniques, the lesson learned from the past is that chemists analyzing wild relatives of sesame should always seek assistance of plant taxonomists specialized in Pedaliaceae to verify the taxonomic assignment of their samples. Species-specific DNA barcodes, which allow reliable taxonomic assignments to non-specialists, are not available in *Sesamum* spp. yet. Because the content of lignans varies greatly among accessions of the same species (e.g., [94], see also Section 2.6), reports of phytochemical analysis should include information about the origin of the samples. Many past studies provided limited or no information about their samples, impeding comparison among studies.

Discoveries of new lignans in wild relatives of cultivated sesame will motivate research about the genes and enzymes involved in their biosynthesis.

## 3. Biosynthesis of Lignans

### 3.1. Biosynthesis of Furofuran Lignans in Sesame

The biosynthesis of lignans in plants has been reviewed [236,237], but no current review focusing on sesame is available. Lignans of sesame and other plant species originate from the oxidative dimerization of two molecules of coniferyl alcohol, which is the central metabolite of the phenylpropanoids pathway [238]. The condensation of coniferyl alcohol yields pinoresinol through a laccase enzyme while the stereoselectivity of the reaction is controlled by a dirigent protein [85,239] (Figure 3). Dirigent proteins are widespread in land plants. They are involved in the dimerization of molecules through radical-radical coupling [240,241]. In the case of pinoresinol formation, the first step consists of the formation of radicals of substrate coniferyl alcohol, which is presumably catalyzed by a laccase enzyme, producing free monolignol radicals (CA·). Two CA· molecules bind to a dirigent protein, which enables stereospecific dimerization to either (+) or (-)-pinoresinol, depending on the plant species [239]. Dirigent proteins form pockets with conserved amino acid residues that control the orientation of the substrate [239]. In the case of *S. indicum*, there is a putative dirigent protein, XP_011080883 [27], in which most of the differing amino acids match those of the (+) enantiomer, in line with the predicted (+)-pinoresinol in this plant species.

Until 2006, it was believed that in *S. indicum*, (+)-pinoresinol was converted to (+)-piperitol and then to (+)-sesamin by two consecutively acting P450 enzymes tentatively designated piperitol synthase and sesamin synthase [242]. However, in 2006 Ono and colleagues showed that in *S. indicum*, the same cytochrome P450 enzyme (CYP81Q1) catalyzes the formation of both, (+)-piperitol and (+)-sesamin, through dual methylenedioxy bridge formation (Figure 9, [85]). Therefore, the enzyme was named piperitol/sesamin synthase (Figure 9, Table 5). In 2019, the structure–function relationship of this enzyme from *S. indicum* was confirmed in a heterologous system, and it was demonstrated that the CYP reductase1 gene product (CPR1) is needed for pinoresinol to sesamin conversion to facilitate electron transfer from NADPH to the CYP81Q1 enzyme [243].

Ono and coworkers [85] found homologs of CYP81Q1 protein in *S. radiatum* (CYP81Q2) and *S. alatum* (CYP81Q3). Recently, it was demonstrated that, unlike CYP81Q1 and CYP81Q2, the CYP81Q3 protein from the wild *Sesamum* species *S. alatum,* produces only a single methylenedioxy bridge and has diastereomeric selectivity, i.e., it can catalyze the formation of (+)-pluviatilol from (+)-epipinoresinol but it accepts neither (+) nor (-)-pinoresinol as a substrate [89]. Due to the absence of CYP81Q1, *S. alatum* cannot synthesize molecules with dual methylenedioxy bridges such as (+)-sesamin and (+)-sesamolin. Instead, this species accumulates furofuran lignans with single methylenedioxy bridges, e.g., (+)-2-episesalatin and (+)-fargesin (Figure 10). The asymmetric configuration of (+)-epipinoresinol is probably the reason for the inability of the CYP81Q3 enzyme to form the second methylenedioxy bridge, and for the hydroxylation or *O*-methylation of the second aromatic ring of (+)-pluviatilol to form (+)-2-episesalatin [89].

Murata and coworkers [246] showed that CYP92B14, a P450 monooxygenase, is responsible for the oxygenation of (+)-sesamin to form (+)-sesamolin or (+)-sesaminol via oxidative rearrangement of an α-oxy-substituted aryl group or direct hydroxylation, respectively (Figure 9). They also demonstrated the functional coordination between CYP92B14 and CYP81Q1 (both P450 enzymes), in which the activity of the former enzyme is enhanced by CYP81Q1.

Sesamin has been suggested to be a precursor of the keto-lignan (+)-episesaminone, another furofuran lignan [86] (Figure 9). Episesaminone was isolated for the first time from unroasted and unbleached sesame seeds as well as from freshly harvested seeds of *S. indicum* [86]. In 2006, a diglucoside of episesaminone, episesaminone-9-*O*-β-d-sophoroside, was found in the perisperm of *S. indicum* seeds [96].

In *S. indicum*, the biosynthesis of furofuran lignans continues from (+)-sesaminol through a series of glucosylation steps (Figure 9). The final product is sesaminol triglucoside (STG), which is a water-soluble lignan that accumulates in high amounts in sesame seeds [102]. The three glucosylation steps of (+)-sesaminol are catalyzed by uridine diphosphate (UDP)-dependent glycosyltransferases (UGTs) (Table 5). The first glucose is always connected to the lignan aglycon via a β-glycosidic bond while the second and third glucose molecules are connected via 1,4- or 1,6-β-glycosidic bonds.

The first two glucosylation steps of (+)-sesaminol were previously identified in *S. indicum* [247]. Only recently, the last missing step was discovered by Ono and coworkers [248]. The first step is the 2-*O*-glycosylation of (+)-sesaminol to produce sesaminol monoglucoside (SMG) by the enzyme UGT71A9. Homologs of UGT71A9 were found in *S. radiatum* (UGT71A10) and *S. alatum* (UGT71A8) [247], evidencing the conservation of these steps in *Sesamum* spp. The glucosylation of SMG leads to diglucoside SDG(β1→6) [(+)-sesaminol 2-*O*-β-d-glucosyl-(1→6)-*O*-β-d-glucoside] or SDG(β1→2) through consecutive glucosylation of the 6′ or 2′ -hydroxyl group of the sugar moiety by the enzyme UGT94D1 or UGT94AG1, respectively (Figure 9) [248]. Until recently, it was not known which glucosylation step of SMG occurs first, β1→6 or β1→2 glucosylation. Ono and coworkers [248] showed that β1→2 followed by β1→6 glucosylation is the major pathway. With this discovery, the missing last step for STG biosynthesis was established. The same authors found an enzyme similar to UGT94D1, called UGT94AA2, which interacts with UGT71A9, UGT94AG1, and, interestingly, also with CYP81Q1 (the piperitol/sesamin synthase) producing, what the authors called, an STG metabolon. In this proposed metabolon for STG biosynthesis, CYP81Q1 acts as a membrane bound protein, which increases the efficiency of sequential glucosylation steps by recruiting three UGT enzymes (UGT71A9, UGT94AA2, and UGT94AG1).

### 3.2. Biosynthesis of Other Lignans in Sesame

Pinoresinol is the precursor of the main furofuran lignans in sesame, i.e., the hydrophobic lignans (+)-sesamin and (+)-sesamolin and the water-soluble STG. However, pinoresinol is also the precursor of dibenzylbutyrolactone class of lignans, which in most plants are produced via reductive cleavage of furofuran rings by pinoresinol–lariciresinol reductases (PLR). These enzymes convert pinoresinol to lariciresinol and then to secoisolariciresinol [245] (Figure 9, Table 5). In sesame, SinPLR1 (XP_011092596) and SinPLR2 (XP_011092597) (Table 5) are putative PLRs; the latter one is similar to a PLR in *Forsythia intermedia* (FiPLR1), which reduces preferably (+)-pinoresinol to (+)-lariciresinol, while SinPLR1 is similar to MgPLR1 in *Mimulus guttatus* [245]. The oxidation of secoisolariciresinol to matairesinol by secoisolariciresinol dehydrogenase (SDH) occurs in some plants such as *F. intermedia.* In this species, the protein SDH_Fi321 converts (−)-secoisolariciresinol to (−)-matairesinol [244] (Figure 10). We have found a gene encoding similar protein in sesame genome (Table 5, XP_011094269, E-value: 4 × 10^−149^, percentage identity: 73.8%), which is a candidate for the homolog of SDH_Fi321. The activity remains to be demonstrated experimentally. Interestingly, the gene was upregulated in mature seeds as compared to young seeds (Andargie, Vinas and Karlovsky, unpublished data).

### 3.3. Biosynthesis of Lignans in Other Plant Species

Lignans are not ubiquitous in the plant kingdom, yet they occur in phylogenetically distant vascular plants [250]. The knowledge of lignan pathways in different plants helps understanding the evolution of lignan biosynthesis, and the genes of lignan biosynthesis from other plants facilitate search for yet unknown genes of the pathway in sesame. Plants in which lignans were characterized include species from the orders Asterales (*Arctium lappa*) [251], Ericales (*Lyonia ovalifolia*) [252], Lamiales (*Sesamum* spp., *Forsythia* spp.) [252,253], Malpighiales (*Linum* spp.) [254], and Malvales (*Wikstroemia sikokiana*) [255]. Biosynthetic studies have been made mainly in lignan-rich plant species of the genera *Sesamum*, *Forsythia* and *Linum.* In the genus *Sesamum*, lignan biosynthesis was studied mainly in *S. indicum*; other lignans have been characterized in *S. alatum* (e.g., 2-episesalatin and fargesin) [89] (Figure 10).

Among the eight classes of lignans [250], the most studied are furofuran, dibenzylbutyrolactone, and arylteralin lignans. All of them are produced from pinoresinol through the same initial step, i.e., oxidative dimerization of coniferyl alcohol (Figure 3, Figure 9 and Figure 10). Furofuran lignans are produced by *Sesamum* spp. (e.g., sesamin, sesamolin, sesaminol and phylligenin) and *Forsythia* (e.g., epipinoresinol and phylligenin) [256,257,258,259]. Dibenzylbutyrolactone lignans are produced also by *Forsythia* spp. (e.g., arctigenin) [256], *Linum* spp. (e.g., yatein) [259] and possibly also by *Sesamum* spp. [245]. *Podophyllum* spp. and *Linum* spp. produce aryltetralin lignans (e.g., podophyllotoxin) [260,261,262].

Phylligenin is produced from pinoresinol via epipinoresinol in *F. intermedia* and *S. alatum* [89,256], whereas 2-episesalatin is produced in *S. alatum* via pluviatilol through consecutive steps of hydroxylation and *O*-methylation [89] (Figure 10). Arctigenin is produced from the *O*-methylation of matairesinol in *F. intermedia* [256,263]. Podophyllotoxin, an aryltetralin lignan with anticancer properties, is produced from matairesinol via pluviatolide in *Linum* and *Podophyllum* species [264] (Figure 10). The cytochrome P450 enzyme responsible for matairesinol to pluviatolide via methylenedioxy bridge formation were identified in *P. hexandrum* (CYP719A23) and *P. peltatum* (CYP719A24) by Marques and collaborators [249]. The next steps from pluviatolide were published for *P. hexandrum* in 2015 by Lau and Sattely [265]. The enzyme *O*-methyltransferase (OMT3) catalyzes the methylation of pluviatolide to form 5′-desmethoxy-yatein, which is then hydroxylated by CYP71CU1 to 5′-desmethyl-yatein. The methylation of this intermediate by enzyme OMT1 produces yatein (a native substrate for ring closure) (Figure 10). The biosynthesis of deoxypodophyllotoxin is catalyzed by the enzyme deoxypodophyllotoxin synthase (2-ODD) from yatein forming the core of the aryltetralin scaffold by oxidative ring closure [265]. The next steps to form podophyllotoxin in *P. hexandrum* have not been identified yet. However, Lau and Sattely [265] demonstrated that two different P450 enzymes (CYP71BE54 and CYP82D61) produce 4-desmethyl-deoxypodophyllotoxin and 4-desmethyl-epipodophyllotoxin (an etoposide lignan) from deoxypodophyllotoxin, respectively (Figure 10). In *L. flavum*, the biosynthesis of 6-methoxypodophyllotoxin (a cytotoxic lignan) occurs through deoxypodophyllotoxin via three steps, including a 6-hydroxylase (DOP6H) enzyme, which is a cytochrome P450-dependent monooxygenase, followed by *O*-methylation by β-peltatin 6-*O*-methyltransferase (βP6OMT) enzyme and 7-hydroxylation with a yet unknown enzyme [262,266]. On the other hand, although the enzyme has not been characterized, hydroxylation of carbon 7 of deoxypodophyllotoxin is the putative step that produces podophyllotoxin in various *Linum* species [261] (Figure 10).

Lignans normally accumulate as glucosides. In *Forsythia* and *Linum*, the enzyme UGT71A18 is responsible for the glucosylation of furofuran lignans [(+)-pinoresinol, (+)-epipinoresinol and (+)-phylligenin] [267], while UGT74S1 is responsible for the glucosylation of the dibenzylbutyrolactone lignan (+)-seicolariciresinol [268]. Enzymes responsible for the glucosylsation of matairesinol and arctigenin are still unknown. In 2016, three *Forsythia* UGTs were suggested for matairesinol glucosylation (CL10456contig1, CL14684contig1 and CL15275contig1) based on molecular analyses of virtual primer-based sequences assembly (VP-seq) [264].

## 4. Genetics of Lignan Synthesis

### 4.1. Genes Involved in Lignan Synthesis in Sesame

The genome of sesame (2n = 26) is composed of 16 linkage groups (LGs). The first two genomes of sesame were sequenced in 2013 [21] and 2014 [27]. Since then, many genomes of *S. indicum* have been sequenced. In large-scale effort to associate agronomic traits with genetic variations, genomes of 705 sesame accessions were re-sequenced [27]. The assembly obtained in 2014 is still used as a reference genome, which is 357 Mb large and contains 27,148 predicted protein-coding genes [27].

Although most enzymes involved in the biosynthesis of furofuran lignans in sesame are known, most of the genes remain uncharacterized. The genes were putatively identified by computational analysis using gene prediction methods (Table 6). These 9 genes are located on 8 LGs. Although secondary metabolite pathways in plants are often scattered [269,270] while only few pathways are encoded by gene clusters [271], the scattering of the lignan pathway in *S. indicum* is remarkable.

The GC content of the exons of these genes varies from 40.6% to 53.6%. After removal of two genes with an unusually low GC content (Acc. Nos. LC199944 and LC484013), the range narrowed to 45.2–53.6%, which is similar to the average GC content of all protein-coding regions in the genome of sesame [26,27]. The low GC content of LC199944 and LC484013 is reflected by the low effective number of codons (*Nc*), which is a measure of codon usage bias [272]. Why is the GC content of these two genes so low? Inspection of putative translation products reveals a high content of amino acids with AT-rich codons. For instance, the protein product of LC484013 has the highest relative content of Phe and Ile, and together with the LC199944 the highest content of Asn and Tyr, among all proteins encoded by the genes listed in Table 6. Therefore, protein composition, resulting from a selection pressure on enzyme function, likely accounted for the unusually low GC content of the two genes.

### 4.2. Expression of Genes of the Lignan Biosynthetic Pathway in Sesame

Most studies of the expression of lignans pathway in sesame used seeds of different maturity. For example, the expression of the genes encoding putative dirigent protein (NCBI geneID = LOC105164033) and for piperitol/sesamin synthase (*CYP81Q1)* was higher in early-mid than in late developmental stages of the seeds (10-20 days post-anthesis) [27]. The expression profile of the *CYP92B14* gene coincided with that of *CYP81Q1*, suggesting a catalytic cooperation between both enzymes, as demonstrated by Murata et al. [246]. These authors also expressed *CYP81Q1*, *CYP92B14,* and *CPR1* simultaneously in a yeast heterologous system. Using (+)-sesamin as a substrate, high amounts of (+)-sesaminol and (+)-sesamolin were produced in the heterologous system expressing only *CYP92B14* and *CPR1*. CYP81Q1 or CYP92B14 alone did not show any catalytic activity. The gene *CPR1*, needed for the enzymatic activity of CYP81Q1, is expressed throughout the entire seed development [246].

Wei and coworkers [26] showed that genes related to oil biosynthesis in sesame seeds were strongly associated with sesamin and sesamolin content. For example, a single nucleotide polymorphism within the gene *SiNST1* (SIN_1005755, NCBI geneID = LOC105173057) defined two alleles, “A” and “C”. The “A” allele was associated with a low content of oil and protein and also with a low content of sesamin and sesamolin, but with a high content of lignin and an accumulation of woody tissue in seeds. The “C” allele was associated with a higher content of oil, protein, sesamin and sesamolin, but with a low content of lignin. The gene was strongly expressed in young seeds [26]. In our work, the gene was expressed in mature seeds to a higher level than in young seeds (Andargie, Vinas and Karlovsky, unpublished data).

Recently, it was shown that sesamin binds steroleosin B, a membrane protein found in oil bodies of sesame seedlings [273]. The gene encoding steroleosin B was expressed throughout the entire development of seeds. *Arabidopsis thaliana* accumulating sesamin and steroleosin B due to the expression of sesame synthase and steroleosin B genes from sesame displayed severe growth defects, indicating that the complex of steroldeosin B and sesamin might interact with signal transduction pathways controlling plant development (see Section 6).

Our own gene expression experiments with young and mature seeds of sesame corroborated that the genes involved in lignan biosynthesis, i.e., *CYP81Q1* catalyzing the synthesis of (+)-sesamin, and *CYP92B14* catalyzing the synthesis of (+)-sesaminol and (+)-sesamolin, were expressed to higher levels in young seeds than in mature seeds. In contrast, the gene SiNST1, which apparently affects partition of phenylpropanoids into lignans and monolignols (see above), was stronger expressed in mature seeds. Similarly, we found that the gene putatively involved in the conversion of secoisolariciresinol into matairesinol (NCBI geneID of LOC105174016) was expressed in mature seeds to a higher level than in young seeds (Andargie, Vinas and Karlovsky, unpublished data). The activation of lignan synthesis at later phases of seed development is also supported by the report that lignans were absent from young seeds of sesame [85].

In line with the focus of lignan research in sesame on the seeds, expression of the lignan pathway in sesame was studied nearly exclusively in seeds. In 2006, Ono and coworkers [85] in their characterization of piperitol/sesamin synthase reported that the gene (designated CYP81Q1, see Table 5 and Figure 9) was expressed in maturing seeds and in leaves but not in leaf petioles or stems. They have not found sesamin or any other lignan in the leaves. The homologue of CYP81Q1 in *Sesamum radiatum*, designated CYP81Q2, was expressed only in seeds [85]. The expression of CYP81Q1 in the leaves of *S. indicum* was confirmed by Hata and coworkers [232], who also detected sesamin in the leaves for the first time. Comparison of two varieties of sesame revealed large differences in the expression of CYP81Q1 and the content of sesamin in the leaves. Expression of CYP81Q1 at a very low level was also detected in the stems of young plants, though no sesamin was detected in these samples [232]. Continuous light for two weeks dramatically increased the expression of *CYP81Q1* in the leaves of sesame, resulting in increased accumulation of lignans in leaves [274]. This finding could be exploited for the commercial production of sesamin from the waste left after threshing, especially if lignan content in seeds can be increased genetically (see Section 7.1).

### 4.3. Molecular Markers and Heritability of Lignan Synthesis

Large differences in lignan content among accessions and varieties (see Section 2.6) indicate the existence of genetic polymorphism, which can be exploited for breeding. For the improvement of sesame as functional food, it is important to understand the heritability of the content of major lignans in sesame seeds. Data on the inheritance of lignan content are limited.

Secondary metabolite patterns in sesame are incongruent with the genetic relatedness [216], similarly to other plants [218,219]. Molecular markers are needed for the prediction of secondary metabolite production. A gene associated with sesamin and sesamolin accumulation was identified in a massive GWAS (genome-wide association study) based on full genomes of 705 sesame varieties, which lead to the identification of genes controlling oil yield [29]. One of these genes, designated *SiNST1*, also controlled lignan synthesis. Seeds of varieties carrying an *SiNST1* allele associated with reduced content of oil, sesamin, and sesamolin, also contained more lignin. This observation was consistent with the function of a homologous gene *NST1* in *Arabidopsis thaliana*, which controls the synthesis of secondary cell wall [275].

In a study from the National Institute of Crop Science in Tsukuba, Japan [212], F_2_ populations derived from accessions with high and low lignan content showed a continuous distribution of sesamin and sesamolin levels, indicating that lignan content was controlled polygenically. The heritability of lignan content was high, hence selection of lines with a high or low lignan content was possible. In a separate study on F_5_ and F_6_ recombinant inbred lines (RILs) from a cross between an accession with high lignan content and a sesamolin deficient accession, Yamamoto [276] showed that the content of sesamin and the sum of sesamin, sesamolin and sesaminol triglucoside were inherited as polygenic traits. In contrast, the content of sesamolin and sesaminol triglucoside were controlled by a single gene and several genes, respectively. A dense set of molecular markers was developed for F_6_ RILs originating from a cross between parents with contrasting metabolic profiles [31]. This material may facilitate further analysis of the *loci* identified by Yamamoto [276]. The first sesame variety bred for high lignan content was reported in 2017 from Japan [277]. The inheritance and general combining ability (GCA) of sesamin and sesamolin content were investigated at Kalasin University, Thailand [185]. Not surprisingly, the inheritance of sesamin and sesamolin content exhibited high combining ability, with both additive and dominant effects controlling the lignan content. Positive correlation between the content of different furofuran lignans (see Section 2.6) indicates that the rise of the content of any lignan will likely lead to an increase of the content of the other lignans. Breeding efforts to enhance lignan content in sesame were recently reported from India [213]. The authors confirmed a high general combining ability of sesamin and sesamolin content. Unfortunately, high-content parents were only crossed with a low-content tester but not with each other; therefore, none of the progeny reached the lignan levels of the best parents. With systematic efforts targeting lignans just starting, breeding for lignan content in sesame is in its infancy.

## 5. Biological Activities of Lignans

### 5.1. Lignans as Health Promoting Agents: From Folk Medicine to Food Additives

It has been recognized for a long time that consumption of sesame benefits health [10,11,42,278]. As early as in 1940’s, injection or consumption of sesame oil was reported to prolong the life span, increase the number of pregnancies, and improve the ability to rear progeny in rats [279,280]. Later studies attributed some of these effects to lignans. Since then, plethora of reports supported the assessment of lignan-rich sesame products as functional food that helps preventing diseases and indicated that some biological activities of lignans could even be used for therapeutic purposes. Although many medical applications of lignans of sesame were protected by patents, to our knowledge no lignan or lignan-based compound has been approved as an active component of a medical product so far.

Antioxidant activity of lignans and tocopherol account for part of the health benefits of sesame consumption (see Section 5.2.1). However, interactions with specific molecular targets and activation or suppression of signal transduction pathways by lignans have also been reported. The latter may be pharmacologically interesting. Suitability of a compound as oral drug can be assessed by predicting its absorption, distribution, metabolism, excretion, and toxicity (so-called ADMET profile) from physicochemical properties such as molecular weight, water solubility, dipole moment, and lipophilicity. Pilkington carried out such a chemometric analysis for lignans [281]. She found that most lignans including the furofuran lignans of sesame did not possess lead-like properties, indicating that their derivatives would not have a good chance to become real-world therapeutics, but they fulfilled the requirements for drug-like compounds. To be potentially useful for the prevention or treatment of diseases, compounds classified as drug-like by chemometry must possess suitable biological activities. The following sections provide ample support for such activities of lignans of sesame.

### 5.2. Biological Activities of Lignans in Mammals and Their Applications

#### 5.2.1. Antioxidative Activity

Antioxidative activity of sesame oil, which likely accounts for a large part of the health benefits of sesame consumption [282], has been studied since the 1950’s [7,283]. These effects are mainly attributed to the antioxidant activities of sesamin, sesamol, and tocopherols. An excellent recent review focusing on medical implications of antioxidative properties of lignans is available [284].

Regarding the antioxidative activity of glycosides and aglycons, sesame lignans can be classified into three groups. Lignans of the first group possess antioxidative activities both in glycosylated or unglycosylated form (sesamin, sesaminol, sesamolinol and pinoresinol). The antioxidant activity of sesamin and sesamolin in vitro is weak since they do not possess phenolic hydroxyl groups [284]. Sesamin protected liver from oxidative damage in vivo [285] though its antioxidative activity in vitro was limited. Elucidation of the mechanism revealed that sesamin was converted to metabolic products that were responsible for the effect [156]. In this process, the methylenedioxyphenyl moieties of sesamin were sequentially opened and demethylated, yielding metabolites with one or two catechol moieties. These metabolites exhibited radical scavenging and antioxidative activities similar to pure catechol [156].

Even though sesamolin has been reported to have no antioxidant activity by Kamal-Eldin and Appelqvist [87], sesamolin can act as an antioxidant in vivo, as demonstrated by Kang et al. [286]. The authors fed rats with a diet containing sesamolin and investigated the metabolism of the lignan as well as its effects on the animals. In addition, Suja et al. [287] showed that sesamolin exhibited stronger effects than sesamin despite having the lowest superoxide-scavenging effect in vitro. Pinoresinol has also been shown to exert moderate anti-oxidative activity on scavenging 2,2-diphenyl-1picrylhydrazyl radical [288]. Pinoresinol also prevented oxidative DNA damage in human mammary epithelial cells [289].

The second group consists of lignans that do not possess antioxidant activity as long as they are glycosylated but can be activated by deglycosylation. These include glucosides of pinoresinol and sesaminol [103]. Common beta-glycosidases are not suitable for the deglycosylation of these conjugated lignans (Section 2.2.2). An example of lignans of this group is sesaminol glucosides, which has not suppressed oxidative stress in hypercholesteromic rabbits [290].

The third group encompasses the products of lignan degradation with high antioxidative activities such as sesamol. Sesamol, which is formed from sesamolin during roasting (see Section 2.2.1), has a higher antioxidant activity than sesamin and sesamolin [291]. The benzodioxole group of sesamol scavenges hydroxyl radicals, presumably producing 1,2,4-trihydroxybenzene (hydroxyquinol) [292]. Antioxidative activity of sesamol was demonstrated in many systems in vitro as well as in vivo. For instance, sesamol inhibited lipid peroxidation in rat liver microsomes [293], and it blocked hydroxyl radical-induced deoxyribose degradation and DNA cleavage [294,295]. It also inhibited the mutagenicity of reactive oxygen species in *Salmonella typhimurium* [296]. The potential of antioxidant properties of sesame lignans for cancer protection is discussed in Section 5.2.3.

In addition to direct antioxidative effects, the consumption of sesamin increased the level of vitamin E, which acts as a physiological antioxidant, probably due to the inhibition of the catabolism of vitamin E via cytochrome P450 [297]. Hanzawa and coworkers confirmed the increase of vitamin E level in rats fed sesamin and showed that the levels of vitamin K (menaquinone) in many organs of rats also significantly increased by feeding a diet with 0.2% sesamin [298]. The inhibition of catabolism of tocopherol by sesamin was demonstrated in human and rat hepatic cells [299]. The mechanism of the inhibition was revealed by experiments showing that human tocopherol-omega-hydroxylase (CYP4F2) was strongly inhibited by sesamin [299].

#### 5.2.2. Estrogenic Effects, Alleviation of Postmenopausal Syndrome, and Antiestrogenic Effects

Phytoestrogens are plant metabolites that exert estrogenic activities [300,301]. Because sesame lignans belong to phytoestrogens, they may alleviate postmenopausal syndrome [302]. This hypothesis was supported by results obtained in rodents with surgically induced menopause fed with sesamol [303]. The advantage of lignans in estrogen replacement therapy as compared to hormones is that they do not increase the clotting risk [304]. Hormos Medical Ltd. (Turku, Finland) sought patent protection for the use of pinoresinol, metairesinol, lariciresinol, and other lignans and their isomers for the prevention or alleviation of postmenopausal syndrome [305].

Enterolignans enterodiol and enterolactone, produced in the digestion track of mammals (see Section 2.2.3), affected gene expression in the liver in a similar way as their precursors matairesinol, pinoresinol and sesamin, and directly activated the estrogen signaling pathway [306]. Using a human breast cancer cell line with a reporter gene fusion for the estrogen response, Pianjing and coworkers [307] demonstrated that enterolactone, sesamin, sesamolin and sesamol possessed estrogenic activities. Sesamol, which is commercially available, possessed the highest activity, even higher than enterolactone.

The estrogenic effects of phytoestrogens may disrupt hormonal balance of the bodies of males, impairing male fertility. To assess this risk, researchers in Lagos, Nigeria, evaluated the chronic reproductive toxicity of sesame lignans in male rats [308]. Surprisingly, treatment with *S. radiatum* extracts increased the testicular weight and sperm count and motility in rats. The authors concluded that lignans of sesame improved the sperm quality in a dose-dependent manner; however, the content of lignans and other metabolites potentially responsible for the effects remained unknown. Studies with pure lignans are needed to confirm the results.

Rather than acting as phytoestrogens, sesame lignans pinoresinol and matairesinol exerted moderate antiestrogen activity in yeast strain expressing estrogen receptor fused to a reporter [309]. Because the concentrations of pinoresinol in sesame oil are lower than the concentrations of estrogenic lignans, and matairesinol only occurs in traces (Table 1), these antiestrogenic effects do not appear relevant for human nutrition.

#### 5.2.3. Sesame Lignans and Cancer

The anticancer activity of sesame lignans has been investigated extensively reviewed by [35]. Antioxidative properties of dietary lignans may reduce the incidence of cancer by inhibiting the production of foodborne carcinogens [310] and possibly also by inactivating reactive oxygen species released in the body during inflammation, which might cause DNA damage leading to mutations and tumorigenesis [311]. Most of this research focused on sesamin (reviewed by Majdalawieh et al. [312]). Anticancer activities of enterolactone have recently been reviewed [313].

The therapeutic potentials of sesamin, sesamolin, sesamol, and enterolactone in cancer treatment have been investigated. In the following, selected results from this voluminous research are presented.

Sesamin provided with food suppressed mammary carcinogenesis in rat [314]. In vitro, sesamin inhibited the proliferation of leukemia cells and colon, prostate, breast, pancreas, and lung cancer cells [37]. The modes of action of sesamin in these effects were apoptosis, interference with signal transduction, and suppression of proliferation, invasion, and angiogenesis [37]. In human squamose carcinoma cells, sesamol exhibited antimetastasis-like effects in vitro (suppression of migration and invasion) [315].

Sesamol exhibited chemoprotective effect in a mouse skin two-stage carcinogenesis model, reducing by 50% skin papillomas induced by a tumor promoter [316]. Sesamol also selectively inhibited proliferation of lung adenocarcinoma in a dose-dependent manner [317].

Sesamolin induced apoptosis in human lymphoid leukemia cells and inhibited the growth of human leukemia cells [318,319]. Apoptosis was also the mode of action of sesamolin on human colorectal cancer cells; apart from the induction of apoptosis, sesamolin also prevented cell invasion by blocking the JAK2/STAT3 pathway involved in growth and proliferation [35].

Sesaminol glucosides inhibited the development of colonic precancerous lesions in vivo and protected rats from induced tumors [320]. Aglycon of sesaminol inhibited several human cancer cell lines in vitro stronger than normal cells, including breast, skin, lung, and colorectal cancer [321]. The strongest inhibition was observed in lung carcinoma cells. Investigating the mode of action, the authors found out that in lung and colorectal cancer cells, sesaminol reduced the level of cell cycle regulator cyclin D1. Identification of the sesaminol-binding proteins (mitochondrial adenine nucleotide translocase) shed light on the mechanisms of overexpression of cyclin D1 in cancer cells and its suppression by sesaminol [321].

Summarizing these results, in vitro and in vivo studies showed that lignans of sesame inhibit the growth of cancer cells by different mechanisms including induction of apoptosis, cell cycle arrest, inhibition of the expression of specific genes, and degradation of regulators overexpressed in cancer cells [36,37,302,312].

#### 5.2.4. Neuroprotection

Positive effects of lignans on neurological and cognitive functions have been reported from animal models as well as from trials with human volunteers. In rats with artificially accelerated senescence, sesame lignans reduced cognitive decline [322]. The authors suggested that long-term consumption of sesame lignans might suppress age-related decline of brain function.

Mental fatigue induced in healthy individuals was suppressed by orally administered mixture of astaxanthin (strongly antioxidant carotenoid pigment) and sesamin [323]. Inhibition of membrane currents through voltage-gated ion channels by sesamol [202] indicated that sesamol could suppress seizures. These results are preliminary to support dietary recommendations, but they will inspire further investigation.

Ischemic stroke may cause neuronal damage due to reduced supply of oxygen and glucose to the brain. Orally administered sesamin exerted significant neuroprotection in mouse model of ischemic stroke, reduced infarction volume, and modulated signaling pathways in mouse brain, resulting in reduced inflammatory and stress markers [324]. In rat, sesamin reduced neurological damage caused by surgical occlusion of cerebral artery and prevented depletion of glutathione and glutathione-depending enzymes [325]. The authors suggested that sesamin may be helpful in stroke therapy. Orally administered sesamin and a mixture of sesamin and sesamolin alleviated the effect of cerebral ischemia in gerbils introduced by surgical occlusion of the right carotid artery by reducing the brain damage caused by hypoxis [326]. In the same study, quenching of nitrogen oxide production by rat microglial cells treated with lipopolysaccharides by sesamin and sesamolin was demonstrated, providing a rationale for the neuroprotective effect of lignans on primary microglia [326].

Protection from oxidative damage (see Section 5.2.1) was suggested to account for the neuroprotective effects of sesame lignans [136]. One of the strongest oxidants in the body is hypochlorous acid, which is produced by oxidation of chloride with hydrogen peroxide catalyzed by myeloperoxidase. Oxidative damage caused by reactive products of myeloperoxidase activity is believed to contribute to a range of neurodegenerative diseases such as Parkinson’s and Alzheimer’s disease and multiple sclerosis [282]. Apart from scavenging reactive oxygen species by lignans (see Section 5.2.1), inhibition of myeloperoxidase [327] may in part account for the neuroprotective effects of sesamol.

#### 5.2.5. Cardioprotection and Reduction of Risk of Cardiovascular Diseases by Lowering Blood Pressure, Lipogenesis, and Cholesterol Level

Cardiovascular diseases belong to the leading causes of death in industrialized countries. Lignans of sesame suppress several physiological factors associated with vascular diseases, atherosclerosis, and heart failure. Furthermore, lignans have been conjectured to protect heart muscle from oxidative damage. In the following, selected results demonstrating the effect of lignans on these factors in experimental animals and humans are collated.

Dietary sesamin stimulated the oxidation of fatty acids and suppressed triacylglycerol synthesis, decreasing lipogenesis in rat [328]. Sesamin also prevented the increase in the serum triacylglycerol level following ethanol consumption in rat [329]. Regarding the effect on the cholesterol level in blood, sesamin reduced the activity of 3-hydroxy-3-methyl-glutaryl- CoA reductase, which is the rate-limiting enzyme of cholesterol synthesis in rats [330]. In hypercholesteromic mice, however, sesamin neither attenuated the elevation of cholesterol level caused by diet containing 0.25% cholesterol nor enhanced the effect of stanol esters, which are plant steroids reducing cholesterol level in blood [331]. In hypertensive rats, feeding 1 g sesamin per kg body weight lowered blood pressure and reduced the tendency to develop thrombosis, suggesting that dietary sesamin might help preventing stroke [332]. Sesamin also reduced cardiac disfunction induced by experimental diabetes in rat model [333], indicating that sesamin consumption may help preventing diabetes-inducing cardiac hypertrophy.

Dietary interventions in humans generated mixed results. A short-term dietary intervention with 60 mg sesamin per day significantly reduced the blood pressure of mildly hypertensive probands [334]. Sesamin also reduced the total serum cholesterol and low-density lipoprotein level in hypercholesterolemic patients [335]. A randomized, placebo-controlled study in overweight men and women provided sesame seed equivalents containing 50 mg sesamin per day has not shown any reduction of blood lipids or blood pressure and the markers of systemic inflammation and lipid peroxidation were not affected, though urinary excretion confirmed that lignans were absorbed and metabolized [331]. On the other hand, administration of sesame to post-menopausal women decreased their serum levels of cholesterol and the level of a precursor of androgens and increased the ratio of tocopherols to cholesterol [302]. Differences in the lignan content in sesame seeds used in these studies may account for the difference in their outcome. The use of purified lignans rather than sesame seeds in intervention studies is advisable.

Elevated levels of myoperoxidase activity are associated with increased risk of cardiovascular diseases, and they worsen the prognosis for cardiovascular patients [336]. Sesamol inhibits myeloperoxidase, which may contribute to the protection of heart muscle from oxidative damage [327]. Sesamin may help protecting consumers from cardiovascular diseases by affecting the nitric oxide signal transduction pathway and preventing dysfunction of blood vessels [337].

#### 5.2.6. Antiaging Effects

With increasing life expectancy, the occurrence of age-related diseases has grown. Anti-aging food supplements [338] promise to increase the life quality of the elderly. Lignans of sesame, purified or in mixtures with other compounds, are sold as anti-aging supplements worldwide. They are part of the lignan market, which is currently valued at US$370 million, and is projected to grow to US$ 590 million in 2027 [339]. Experimental support for antiaging effects of sesame lignans have been obtained in invertebrate models and in rodents.

Among the lignans of sesame, sesamin has been investigated most often for antiaging effects. Sesamin extended the life span in the fruit fly *Drosophila melanogaster* [340] and the nematode *Caenorhabditis elegans* [341,342]. In *D. melanogaster* mutant depleted of superoxide dismutase, which exhibits a shortened lifespan and accelerated aging-related phenotype, sesamin extended the life span, suppressed the decline of motoric activities, and delayed aging-related loss of dopaminergic neurons in the brain [343].

In a mice senescence model, induced by chronic subcutaneous injections of D-galactose, oral administration of sesamin suppressed the decline of cognitive and motor capabilities, the loss of body weight, the changes in liver enzyme activities, and the increase of liver malondialdehyde and other markers of oxidative stress [344]. The suppression of age-related cognitive decline by oral administration of sesame lignans was independently confirmed in a different senescent-accelerated mice strain [322]. In mice, sesamin also suppressed age-related disorders of the kidney [345].

Because sesamol formulation prevented photodamage of skin (lesions and ulcers) due to chronic UV exposure [346], the authors speculated that sesamol exerts antiaging effects and protects skin from age-related wrinkling. The use of sesamol in anti-aging skin lotions is protected by a patent owned by the South Korean cosmetics and health care company Amorepacific Group [15]. The company Johnson & Johnson obtained patent protection for the use of paulownin, which is an isomer of sesaminol, in skin care products in the U.S.A. Loy and coworkers [347] are seeking protection for the same invention in Europe [348]. A Japanese research group associated with Kiyomoto Corporation sought protection for the use of sesaminol to stimulate collagen and elastin production and migration of keratinocytes [20]. These activities slow skin aging and promote would healing (see below).

#### 5.2.7. Pain Relieve, Anti inflammatory Effects and Wound Healing

Topical application of sesame oil improved wound healing in rats [349]. Among the components of sesame potentially responsible for the wound healing effect, sesamol was most often studied (e.g., [350]). Composite cellulose acetate-protein membranes impregnated with sesamol were recently designed for the acceleration of wound healing [351]. Membranes loaded with 5% sesamol promoted wound healing and inhibited wound inflammation in diabetic mouse.

Sesaminol stimulates elastin and collagen production in the skin, and the use of sesaminol in skin care products was protected by a patent [20] but these activities also promote wound healing. External use of sesaminol to accelerate wound healing is covered by additional claims of the cited patent.

A number of reports documented anti inflammatory activities of lignans. Sesamin and sesamolin inhibited 5-desaturase activity and caused accumulation of dihomo-γ-linoleic acid, which is a precursor of prostaglandins [352]. In an extensive in vivo investigation on rat and mice, orally administered sesamin inhibited inflammation; reduced edemas induced by carrageenan injections, and suppressed response to pain, indicating analgesic effects [353]. Sesamin also suppressed inflammation of mice retina in a model of diabetic retinal injury [354]. The use of sesamol as anti-inflammation therapeutics was protected by a patent [14].

#### 5.2.8. Miscillaneous Activities: Hepatoprotection, Hypoglucemic Effect, Anti-Osteoporesis Effects, Protection of Cartilage, and Alleviation of Alcohol Sickness

Not all known biological activities of lignans with potential for disease prevention or therapy are covered in Section 5.2.1, Section 5.2.2, Section 5.2.3, Section 5.2.4, Section 5.2.5, Section 5.2.6, Section 5.2.7. Hepatoprotection is the first among these activities. Sesamin protected liver of rodents against damage caused by alcohol or organic solvents [329]. The hepatoprotective effect of sesamin has also been demonstrated in mice treated with carbon tetrachloride [355]. Pinoresinol reduced oxidative damage in liver cells in vitro [288], and a mixture of lignans extracted with methanol from sesame oil reduced oxidative damage of the liver in rats treated with bisphenol A [356].

Inhibitors of α-glucosidase can help reduce blood sugar levels (hyperglycemia) in diabetes mellitus patients after consumption of starchy food. In a Korean study, pinoresinol diglucoside was identified as the major inhibitor of α-glucosidase in leave extracts of hardy kiwi (*Actinidia arguta* (Siebold. & Zucc.) Planch. ex Miq.) [357]. As elaborated above (Section 2.1.4), sesame contains three isomers of pinoresinol diglucoside, only one of which, namely pinoresinol di-*O*-β-d-glucoside, was found in hardy kiwi. Wikul and coworkers showed that (+)-pinoresinol aglycone, as well as sesamin and sesamolin, inhibited α-glucosidase from baker’s yeast [181]. Only pinoresinol, however, efficiently inhibited α-glucosidase (maltase) from rat intestine, indicating that pinoresinol-containing sesame products may support the therapy of diabetes mellitus by reducing blood sugar level [181]. Semisynthetic derivatives of furofuran lignans of sesame were tested for the inhibition of three α-glucosidases and binding of the most potent derivatives to intestinal maltase was elucidated by molecular docking [358].

Pinoresinol diglucoside alleviated osteoporosis in rats [107]; unfortunately, only one of the three isomers was tested (see Section 2.1.4). Sesamol can protect cartilage against degradation (chondroprotective effect) [359]. The use of sesame lignans administered in the form of oil/water emulsion to prevent sickness from drinking alcohol beverages was protected by patents in the U.S.A. [19] and other countries [16].

A number of further activities potentially useful for the prevention or treatment of diseases have been reported. We refer to the excellent reviews by Jeng and Hou [352], Namiki [11], Wu and coworkers [360], and Rodriguez-Garcia and coworkers [42] for further examples.

#### 5.2.9. Harmful Effects of Sesame Lignans in Humans

Induction of squamous cell carcinoma in the forestomach of rats by sesamol has been reported [361]. These results, however, do not seem relevant for public health because high doses of sesamol were administered for a long time, and also because humans do not possess a forestomach [362].

Sesame seeds and oil may cause food allergy. Therefore, labeling of their presence in food products is mandatory in the EU [363]. Allergies caused by sesame oil and seeds accounted for only 1% of food allergies in 1996 [364], but their occurrence grew continuously during the last decades [365]. Food and cosmetics containing sesame oil were reported to cause skin hypersensitivity and urticaria [39]. Apart from albumins and globulins, the lignans sesamin, sesamolin, and sesamol were found to be responsible for allergic reactions [366,367]. As with other food commodities, allergenic potential of sesame only manifests in consumers with an allergic disease of a matching specificity. Therefore, it cannot be regarded as a general foodborne health risk. Certain health effects of sesame lignans actually suppress allergic reactions; for instance, sesamin attenuated allergic inflammation in an asthma model in rat [368].

Cytotoxic products of the oxidation of sesamol by gaseous oxygen have been described [128], but these transformations are unlikely to occur during storage or processing of sesame oil.

Sesamin and episesamin were investigated for genotoxicity (Ames test), chromosomal aberrations in mammalian cell cultures, and effects on DNA and chromatin in vivo [369]. Both lignans were not genotoxic. In all other tests, episesamin was also negative, whereas sesamin at high concentrations caused chromosomal aberrations in vitro. These effects were not confirmed in vivo after oral administration of sesamin up to 2 g/kg, indicating that the consumption of sesamin is not a food safety risk [369].

Many lignans exert estrogenic effects, which may be useful for estrogen replacement therapy (see Section 5.2.2). On the other hand, phytoestrogens might negatively affect human health as endocrine disruptors [304,370]. Furthermore, pinoresinol and matairesinol exerted moderate antiestrogen activity rather than acting as phytoestrogens in a yeast strain containing estrogen receptor and a reporter gene under control of an estrogen-responsive element [309]. Molecular docking revealed that the difference between estrogenic lignans, such as enterodiol, and antiestrogenic lignans pinoresinol and matairesinol might be explained by the ability of different lignans to bind to and stabilize different forms of the receptor. Because the concentration of pinoresinol in sesame oil is lower than the concentrations of estrogenic lignans, and matairesinol occurs only in traces (Table 1), the antiestrogenic effects of pinoresinol and matairesinol do not appear relevant for human nutrition.

### 5.3. Antimicrobial Effects—Application and Therapeutic Potential

Kumar and Singh [207] investigated the growth inhibition of three bacteria by sesamin, sesamolin, and sesamol. Sesamol at 2 mg/mL blocked the growth of *Bacillus cereus* and *Staphylococcus aureus,* but not *Pseudomonas aeruginosa,* while sesamin and sesamol at a concentration of 2 mg/mL have not reached the minimal inhibitory concentration (MIC). Based on these results, the authors suggested that sesamol could be used as an antimicrobial agent against food borne pathogens. We regard the reported MIC value as too high for practical applications.

In other studies, sesamin has not inhibited the growth of *Escherichia coli,* and it inhibited only moderately *S. aureus* [371]. Sesamin has not inhibited the growth of *Mycobacterium tuberculosis* [372]. Bussey and coworkers [373] reported low MIC values, ranging from 8 to 150 µg/mL, for sesamin in nontuberculous mycobacteria. Their results suggested that sesamin and asarinin (episesamin) likely accounted for the medical effects of roots of *Anemopsis californica*, which is used in the folk medicine of North American tribes, and which contains high levels of both lignans. Sesamin and episesamin might thus have potential for the treatment of infections with nontuberculous bacteria.

Hwang and coworkers [374] reported toxicity of (+)-pinoresinol to *Candida albicans*, *Trichosporon beigelii*, and *Malassezia furfur* with MIC values of 12-25 µg/mL. Kulik and coworkers studied the effects of pinoresinol and secoisolariciresinol on the plant pathogenic fungus *Fusarium graminearum* [375]. They have not determined MIC values but reported that both lignans inhibited the growth of the fungus by 5–45% (depending on the strain) at a concentration of 5 µg/mL. More interestingly, pinoresinol at this concentration suppressed the transcription of the biosynthesis pathway for trichothecene mycotoxins nivalenol and deoxynivalenol, and it strongly reduced the level of mycotoxins accumulating in culture medium [375].

Sesamol, which is a degradation product of sesamolin, inhibited lipid metabolism and growth of the zygomycetous fungus *Mucor circinelloides,* but the growth inhibition was moderate, amounting to 10% at 1.5 mM sesamol [376]. At a concentration of 7 mM, sesamol completely inhibited the growth of several basidiomycetous and ascomycetous fungi and yeast [376]. This concentration (ca. 1 mg/mL) is too high for practical applications. Similarly, sesamol at a concentration of 2.5 mM inhibited lipid metabolism and growth in the plant pathogenic fungus *Fusarium verticillioides* [377]. For a comparison, the MIC of a weakly fungicidal preservative sorbic acid was reported 2 mM for mycelial growth of *Aspergillus niger* [378] and 2.7 to 5.4 mM for *Penicillium verrucosum* and *Aspergillus westerdijkiae* [379]. Antifungal plant metabolites inhibit fungal growth at concentrations that are two orders of magnitude lower [380].

To summarize, sesame lignans do not seem promising as general antibiotics or antimycotics, but they may be developed for special niches, such as dimorphic human-pathogenic fungi (demonstrated effect of pinoresinol) and nontuberculous bacteria (demonstrated effect of sesamin). Furthermore, products based on sesame lignans may prevent contamination of food commodities with trichothecene mycotoxins (demonstrated effect on trichothecene synthesis in *F. graminearum*).

### 5.4. Biological Activites of Lignans in Insects: Synergy with Insecticides and Antifeedant Effects

In a search for new insecticides, strengthening of the activity of pyrethrum and rotenone insecticides by sesame oil was discovered, immediately introduced into practice, and protected by a patent [381]. Sesamin was identified as the oil component responsible for the synergistic effect in the USA and England at the same time [382,383], but both labs noticed that sesame oil depleted of sesamin was still active and speculated that additional component(s) contributed to the effect. Soon after, (+)-episesamin (called isosesamin) and asarinin (which is (-)-episesamin) were shown to enhance the effect of insecticides in an extent comparable to sesamin [382]. Interestingly, pinoresinol did not exert any synergistic activity. The use of sesamin as a synergist for insecticides was covered by a patent [384] and the search for further active components continued. The next synergist identified was sesamolin, which turned out to be even more potent than sesamin [385].

The supposed mode of action of natural as well as synthetic synergists is the inhibition of detoxification of insecticides. In a recent study on *Aedes aegypti,* several essential oils including sesame oil and black pepper oil inhibited P450s and potentiated the insecticidal activity of insecticide carbaryl [386].

Apart from synergizing with insecticides, sesame lignans adversely affect insects as antifeedants and as analogues of juvenile hormone, interfering with insect development. Pinoresinol but not sesamol prevented molting of the blood-sucking insect *Rhodnius prolixus* while both pinoresinol and sesamol exerted antifeedant effects on this triatomine [387,388]. Sesamolin inhibited metamorphosis in the milkweed bug *Oncopeltus fasciatus* [389] but the activity was weak, compared with synthetic synergists sesamex (sesoxane) and piperonyl butoxide.

Antifeedant activity is common to many lignans with methylenedioxy-benzene moieties [390]. Sesamin blocked feeding of the caterpillar *Spilarctia obliqua* [391] and inhibited growth of the silkworm *Bombys mori* [392]. Asarinin, which is identical with (-)-episesamin, exhibited strong antifeedant activity against the flour beetle *Tribolium castaneum* [393] with a very low EC50 of 25 mg/kg. Paulownin, which is 8-hydroxy-sesamin, inhibited the growth of insect larvae [394].

Lopseed (*Phryma leptostachya*) contains fufofuran lignans that are similar to lignans of sesame because they contain two methylenedioxobenzyl groups, some of which are methoxylated [395]. One of these lignans (haedoxan A), which contained a benzodioxolan moiety inserted between the furofuran kernel and one of the methylenedioxobenzenes, exhibited very strong stomach toxicity as well as topical toxicity in two pests. The activity of the other lignans, which were more similar to furofuran lignans of sesame, was however weak [395].

Today, sesame oil is a component of the commercial insecticide Organocide (Organic Laboratories, Inc., Stuart, FL, USA), but the mode of action is physical rather than chemical [396].

## 6. Conceivable Biological Functions of Lignans in Sesame Plants

### 6.1. Resistance against Diseases

In spite of the large amount of data on biological activities of sesame lignans, our understanding of their biological functions is limited. Lignans are derived from phenylpropanoids. Plant phenolics contribute to resistance against diseases in many pathosystem [397,398,399]. As will be shown in the following, this cannot be generalized to lignans.

We are aware of only one report indicating that lignans biosynthesis was involved in plant defense against a pathogen: infection of cotton with *Fusarium oxysporum* stimulated the transcription of genes involved in lignan biosynthesis [400]. Extracts of several tissues of sesame inhibited the growth of fungal pathogens of sesame [401], but the chemical composition of the extracts was not examined. The results obtained in other plants do not support a role of lignans in disease resistance. In flax, silencing the cinnamyl alcohol dehydrogenase, which shifted more cinnamyl alcohol to the synthesis of lignans while reducing the content of lignin, slightly decreased the resistance of flax to its major pathogen *Fusarium oxysporum* [402]. Similarly, silencing cinnamyl alcohol dehydrogenase in poplar did not affect the colonization with mycorrhiza [403]. Infection of poplar with fungal pathogens actually induced the synthesis of cinnamyl alcohol dehydrogenase, which diverts the substrate from lignans to monolignols and lignin [404]. Likewise, elicitation of flax cell culture with fungal pathogens did not increase lignan levels in the cells [405].

Some lignans were reported to inhibit dimorphic yeasts and zygomycetous, ascomycetous and basidiomycetous fungi including plant pathogens (see Section 5.3), but except for dimorphic yeasts, the MIC values were too high to indicate a defense function. The observation that low concentrations of pinoresinol inhibited the synthesis of trichothecenes in *F. graminearum* is interesting because trichothecenes are virulence factors facilitating colonization of the host plant; however, *F. graminearum* does not infect sesame. Taken together, the available data do not support the hypothesis that lignans in sesame are resistance factors protecting plants from microbial pathogens.

### 6.2. Resistance against Herbivores

For a long time, sesame oil has been known to increase the efficacy of certain insecticides (see Section 5.4). Insecticidal and antifeedant activities of purified sesame lignans and compounds similar to lignans indicate that lignans of sesame might contribute to the protection of seeds against herbivores. Older literature on antifeedant activities of lignans was reviewed by Harmatha and Dinan [406]. Insecticidal activity was demonstrated for furofuran lignans from Himalayan shrub *Phryma leptostachya*; the structure of these metabolites is very similar to the structure of lignans in sesame [395]. Natural benzodioxole derivatives, reminiscent of 1,2-methylenedioxybenzene moieties of lignans, exhibited insecticidal activity towards leaf-cutting ants [407]. Further support for a role of sesame lignans in defense against herbivores was provided by demonstrating antifeedant activity of sesamin [391] and by the observation that pinoresinol ingested by caterpillars protected the larvae from insectivorous ants [408]. Pinoresinol secreted by glandular hairs on the back of the larvae deterred ants, and pinoresinol-treated flies lost their attractiveness as food for ants.

Another indication that lignans may be involved in plant defense against herbivores is the induction of lignan biosynthetic pathway by methyl jasmonate [409], which is a phytohormone triggering plant defense against insects [410]. These results and antifeedant activity of lignans observed in vitro corroborate the hypothesis that lignans protect sesame seeds from herbivores. The results obtained in vitro should, however, not be overrated. Experiences from other systems show that insecticidal activities of natural products observed in vitro may misguide research into a wrong direction for decades [411]. Genetic manipulation of lignan synthesis in sesame will be necessary to determine the ecological function of lignans conclusively.

### 6.3. Protection from Radiation Damage

UV light induced the synthesis of plant metabolites protecting the plant from radiation damage [412], therefore exposure to light of controlled spectral composition was studied as a means of enhancing the content of desirable secondary metabolites in food plants [413]. Exposure of sesame plants to continuous light stimulated the expression of sesamin synthase gene and the accumulation of sesamin in leaves [274]. In this context, it might be interesting to determine the effect of UV light on the content of sesamin in sesame leaves. The low content of sesamin in the leaves, however, and the absence of lignans from all other organs except seeds and roots (see Section 2.7) render the role of lignans in the protection of plants against UV damage unlikely.

### 6.4. Developmental Control in Seeds

The last potential biological function of lignans is the most speculative: sesamin might be linked to plant development. A recent report [273] showed that sesamin and sesamin-binding protein steroleosin B from sesame seeds caused various developmental defects when concomitantly produced in genetically engineered *A. thaliana* plants [273]. If steroleosin B and sesamin are involved in the developmental control in sesame, they must exert their function in seeds, because steroleosin B is produced in seeds and during germination [273] while sesamin accumulates to high level in mature seeds but disappears immediatly after germination (Figure 1C in [85]). These results are especially interesting in the light of a previous report that sesamin and sesamolin, depending on the enantiomer and concentration, stimulated or inhibited the growth of lettuce and ryegrass [414]. Are lignans involved in signal transduction in plant development? A conclusive answer can only be obtained by comparing isogenic lines of *Sesamum* sp. with genetically manipulated biosynthetic pathways for lignans. This sort of a proof is a standard requirement for the demonstration of biological functions of secondary metabolites in fungi (e.g., [411,415]). Time is mature to apply the same scrutiny to the elucidation of biological functions of plant secondary metabolites. In sesame, the technology is ready (see Section 7.1).

## 7. Outlook: Biotechnology in Lignan Production

### 7.1. Prospects for Genetic Engineering of Sesame Plants

Enhancing agronomic performance is the main goal of genetic engineering of crop plants [416]. The improvement of plant secondary metabolism, however, poses a number of challenges. Secondary metabolites vary from species to species [417] and transferring pathways among plants requires knowledge of the involved enzymes and genes, which are often scattered over several chromosomes (see Table 6) while related activities are catalyzed by similar enzymes [418]. Since secondary pathways are often interconnected via common precursors and/or control elements, modifying the activity of one pathway can affect other pathways [419]. For instance, silencing the gene encoding an enzyme converting pinoresinol into lariciresinol in flax caused increased accumulation of pinoresinol and its glycosides [420]. Analysis of the genes of secondary metabolism and their transcription has been more straightforward than the investigation of enzymatic activities of proteins encoded by these genes, but progress in metabolomics is closing the gap [421]. Concerted efforts during the last decade to unravel the genetic basis underlying agronomically relevant physiological and morphological traits in sesame generated a rich toolbox of genomic resources [21,22,23,25,222,422,423,424]. These resources can facilitate metabolic engineering in sesame.

Heterologous expression of two genes of the lignan pathway of sesame allowed the conversion of pinoresinol to sesamin in *E. coli* [243] and the production of sesamin in *Arabidopsis thaliana* [273]. In roots and seedlings of *A. thaliana* expressing the sesamin synthase gene, the content of sesamin reached 10–80 ng/mg dry weight [273], which corresponds to 0.5–10% of the content of sesamin in sesame seeds (cf. Table 1). Experiences from other systems indicate that optimization of growth conditions and stepwise genetic improvements will likely be needed for high-performance production systems for lignans [425]. Cloning, characterization and manipulation of auxiliary enzymes such as UDP-glycosyltransferases will allow fine-tuning towards practical needs. Genes encoding UDP-lignan-glysosyltransferases from *S. indicum*, *S. alatum*, and *S. radiaum* have been characterized with the explicit aim to facilitate engineering lignan biosynthesis [426].

The perspectives of engineering lignan biosynthesis have been reviewed [427,428]. Successful heterologous expression of several enzymes of the lignan pathway raises hope that manipulation of lignan biosynthesis in sesame becomes possible soon. Apart from heterologous expression in *Arabidopsis* (see above), an encouraging result was achieved in wheat. Expression of pinoresinol lariciresinol reductase from *Forsythia* under a strong maize promoter caused an increase in the secoisolariciresinol diglucoside level by a factor of 2.2 [429]. We suppose that the effect might have been even stronger than reported. Firstly, only the level of diglucoside was determined, rather than the total level of the aglycon in the tissue. If glucosylation was a rate-limiting step in the synthesis of secoisolariciresinol diglucoside in manipulated wheat plants, monitoring secoisolariciresinol diglucoside underrated the level of secoisolariciresinol. Additionally, the analytes were extracted into diethyl ether, which does not appear suitable for the extraction of diglucosides due to its low polarity. Polar solvents used for the extraction of secoisolariciresinol diglucoside used so far were dioxane/ethanol [430], methanol/water, and acetone/water [431]. If the solubility of secoisolariciresinol diglucoside in diethyl ether was limited, which we assume was the case, the relative recovery of the analyte was lower in samples with higher analyte levels, leading to an underestimation of the differences between samples.

Apart from overexpressing genes of metabolic pathways and fine-tuning genetic pathways by genome editing, suppression of undesirable branches of biosynthetic pathways can be achieved by RNA interference (RNAi). To manipulate metabolic pathways by RNAi, double-stranded RNA is produced in the cell to initiate degradation of undesirable mRNAs [432]. RNAi technology was applied to downregulate *LuPLR1* gene in flax (*L. usitatissimum*) seeds. This gene encodes pinoresinol lariciresinol reductase, which is responsible for the synthesis of secoisolariciresinol, the major lignan accumulating in the seed coat of flax seeds [420]. The precursor of secoisolariciresinol and the substrate for the silenced enzyme was pinoresinol, which accumulated in the engineered seeds to a higher level than in non-modified seeds. Furthermore, new lignans not found in wild type flax were identified in the transgenic seeds, indicating the complexity and potential pitfalls but also the potentials of genetic engineering of the lignan pathway. *Forsythia* suspension cells were engineered for increase pinoresinol production by blocking the activity of pinoresinol/lariciresinol reductase. Cells transfected with a suitable RNAi construct exhibited a complete loss of matairesinol but accumulated 20-fold higher amounts of pinoresinol and its glucoside when compared to non-transformed cells [433].

Manipulation of transcription factors controlling phenylpropanoid metabolism [434,435] and the lignan pathway [436] is another option for breeders to improve the content of lignans in sesame seeds. The fact that transcription factors controlling phenylpropanoid metabolism sometimes recognize their targets in other plant species [437] extended the toolbox for engineering lignan synthesis.

The first attempt to transform *S. indicum* using *Agrobacterium tumefaciens* was undertaken in 1999 [438] but the difficulty of regenerating sesame plants from callus hampered progress in the field. Careful selection of accessions amenable to transformation and the optimization of regeneration conditions eventually allowed for the development of a reliable transformation protocols with an acceptable transformation efficiency [439,440,441]. A successful particle bombardment protocol for sesame was also developed [442]. Genetic transformation of sesame is therefore accessible.

The fact that genes encoding enzymes involved in the synthesis and transformations of lignans are being increasingly protected by patents indicates that the industry seriously pursues genetic engineering of lignan synthesis in sesame [18,426,443].

### 7.2. Cell and Hairy Root Cultures for Lignan Production

Lignans can be produced in cell cultures [444] and hairy root cultures [445]. Hairy root cultures are particularly attractive for genetic engineering of secondary metabolism in crops recalcitrant to genetic transformation. Hairy roots are adventitious roots genetically transformed with *Agrobacterium rhizogenes,* which harbors the root inducing (Ri) plasmid and *rol* (root loci) genes that trigger hairy root growth. Compared to *A. tumefaciens*-mediated genetic transformation, *A. rhizogenes* mediated transformation is more efficient; no regeneration of transformed tissue is needed; and the transformed organ (root) proliferates fast. The fact that cell and hairy root cultures can be grown under controlled conditions independently of the season and climate over the whole year [446] compensates for the high costs of facilities for in vitro production, as compared to growing plants in the field. For instance, sesame seeds contain 100 times more sesamin than transgenic cells of *Forsythia*; however, transgenic cells multiplied 10-times in two weeks [433], and they can be cultivated without seasonal interruptions. External factors such as light can be optimized in order to maximize sesamin production in cell cultures [447,448]. Culturing conditions exert tremendous effect on the metabolic productivity of tissues and cells. For instance, the same hairy root culture of *S. indicum* accumulated certain metabolites in a bioreactor, while these metabolites were essentially missing when the roots were grown in flasks [223].

Hairy roots are believed to possess great potential for the production of alkaloids of medical plants [449]. Hairy roots of *Linum flavum* produced 2 to 5 times more 6-methoxypodophyllotoxin than non-transformed roots and 5 to 12 times more than cell suspensions [450]. Hairy root cultures of *Linum album* accumulated 40–58 mg/g of 6-methoxypodophyllotoxin, 0.29–2.28 mg/g *β*-peltatin and 0.20–0.25 mg/g podophyllotoxin [451]. Hairy roots of *L. austriacum* accumulated 16.9 mg/g dry weight of justicidin B while cell cultures accumulated just 6.7 mg/g [452]. Ogasawara and coworkers reported that hairy roots from *S. indicum* accumulated sesamin and sesamolin [226], but Kareem found only sesamin in hairy root cultures of 25 accessions of *S. indicum* [223].

Genetic engineering of metabolic pathways can be achieved in the course of the initiation of hairy root cultures because the process is based on genetic transformation. As *A. tumefaciens* and *A. rhizogenes* use essentially the same mechanism for the transfer of DNA into plant cells and its integration into plant genome [453], binary vectors developed for *A. tumefaciens* can be used with *A. rhizogenes*, too [454]. Transformation vectors derived from the root-inducing megaplasmid of *R. rhizogenes* are available [455]. Genetic manipulation of lignan synthesis in hairy roots has not been attempted in sesame yet, but it has been achieved in another plant. The content of lariciresinol in hairy roots of Chinese medical plant *Isatis indigoticas* increased from 24 mg/g to 96 mg/g by overexpression of a single gene of the lignan pathway (the gene IiC3H encoding P450 hydroxylase; [409]). Because hairy root cultures of *S. indicum* can be generated easily (e.g., [223,456]) and many genes of the lignan pathway have been characterized (see Section 4.1), genetically modified hairy roots appear to be a feasible approach towards engineering lignan synthesis in sesame.

## Figures and Tables

**Figure 1 molecules-26-00883-f001:**
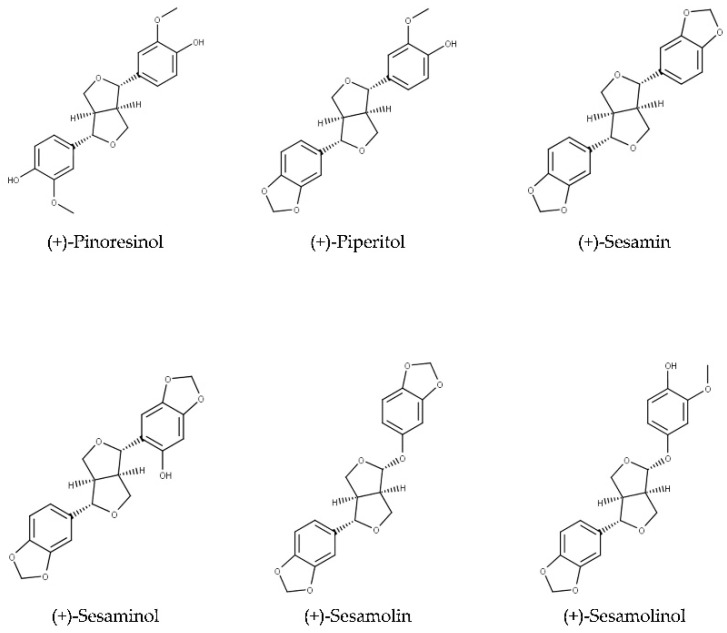
Structures of major lignans of sesame.

**Figure 2 molecules-26-00883-f002:**
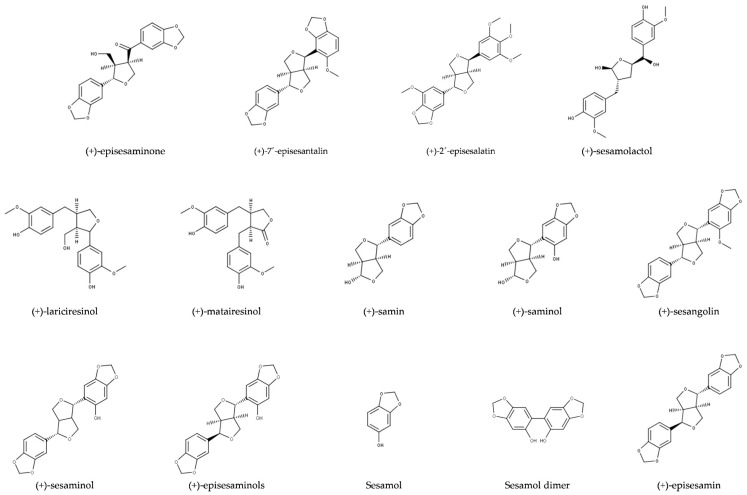
Minor lignans of sesame (first two rows) and transformation products (bottom row).

**Figure 3 molecules-26-00883-f003:**
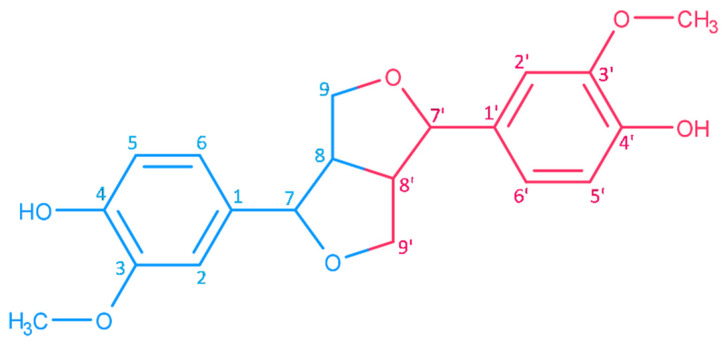
Dimerization of cinnamyl alcohol forms pinoresinol. Numbering according to IUPAC recommendation [34].

**Figure 4 molecules-26-00883-f004:**
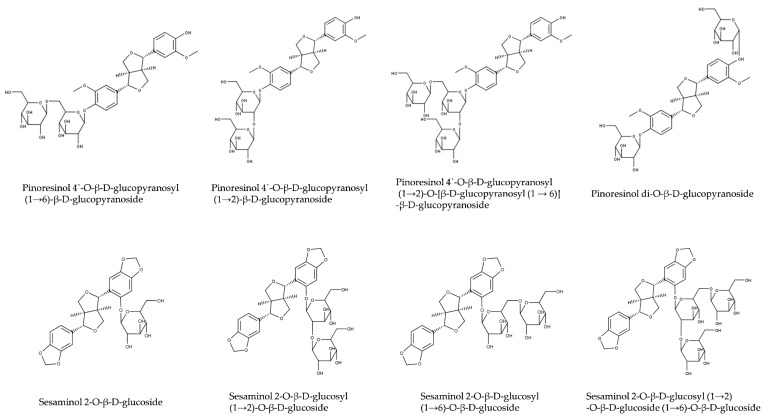
Structures of major lignan glycosides of sesame.

**Figure 5 molecules-26-00883-f005:**
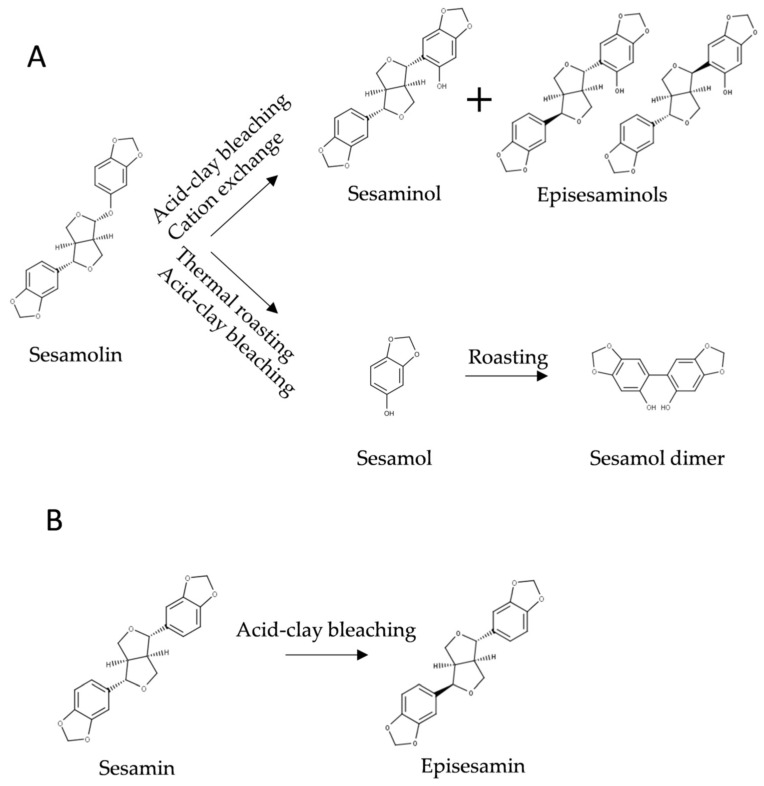
Degradation and transformation products of sesame lignans during industrial processing. (**A**) Degradation products of sesamolin. (**B**) Degradation products of sesamin.

**Figure 6 molecules-26-00883-f006:**
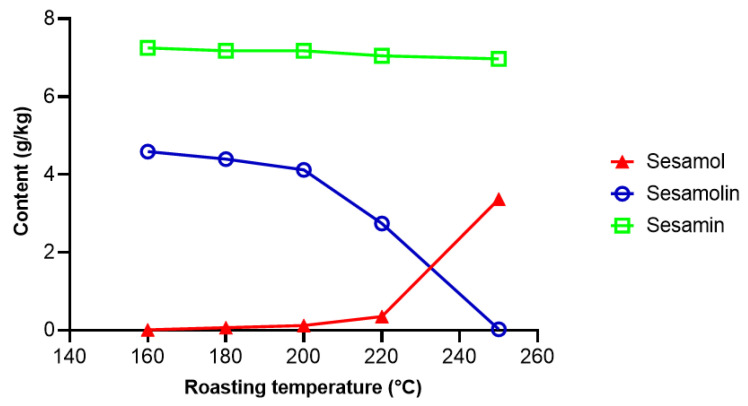
Conversion of sesamolin into sesamol by roasting. Seeds were roasted for 25 min in an electric oven at a designated temperature. The graph was constructed using data published by Yoshida and Tagaki [126].

**Figure 7 molecules-26-00883-f007:**
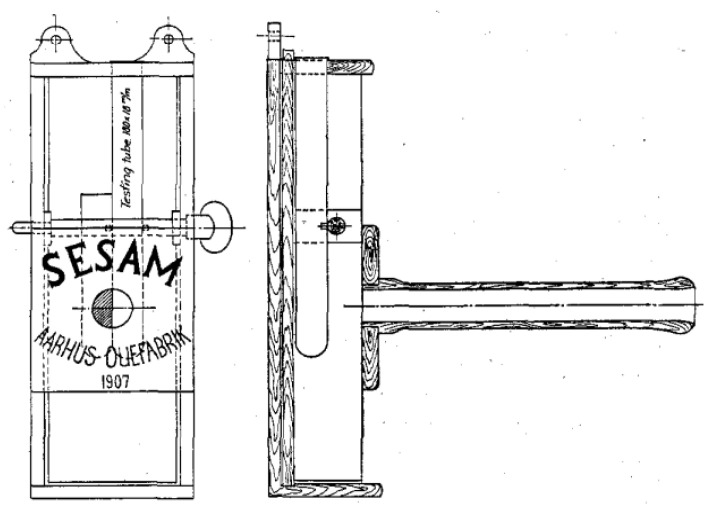
Colorimeter for Villavecchia test used in Aarhus Oliefabrik (Aarhus, Denmark).

**Figure 8 molecules-26-00883-f008:**
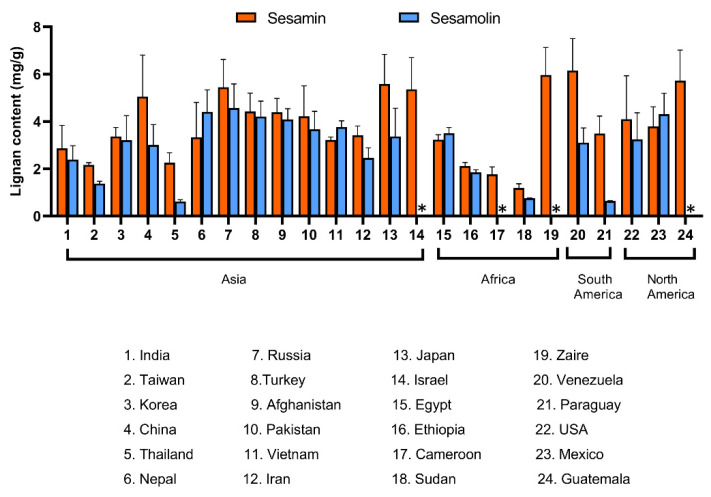
Variation of lignan content in sesame seeds. Lignan content is given in mg/g. Asterisk indicates that sesamolin content was not determined.

**Figure 9 molecules-26-00883-f009:**
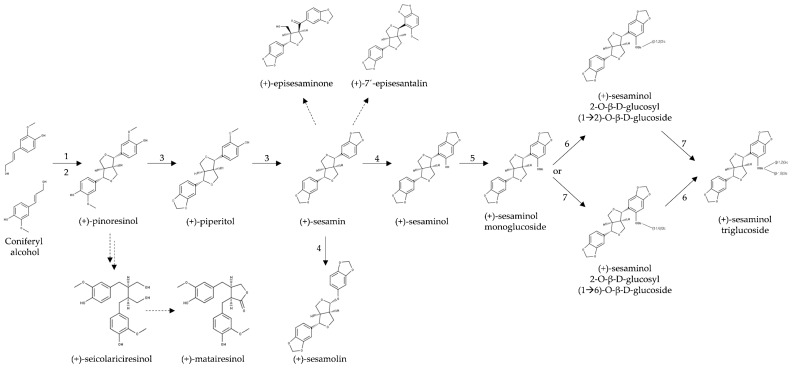
Lignan biosynthetic pathway in *Sesamum indicum*. 1 = laccase, 2 = dirigent protein, 3 = piperitol/sesamin synthase (CYP81Q1), 4 = sesamolin/sesaminol synthase (CYP92B14), 5 = UGT71A9, 6 = UGT94AG1, 7 = UGT94D1/UGT94AA2. Identified and postulated conversions are represented by solid and broken lines, respectively. The sequential conversion from pinoresinol to secoisolariciresinol and to matairesinol is proposed in sesame based on *Forsythia intermedia* biosynthetic pathway [244,245]. The convertion of sesamin to 7′-episesantalin has been recently proposed to occur in *S. radiatum* [92].

**Figure 10 molecules-26-00883-f010:**
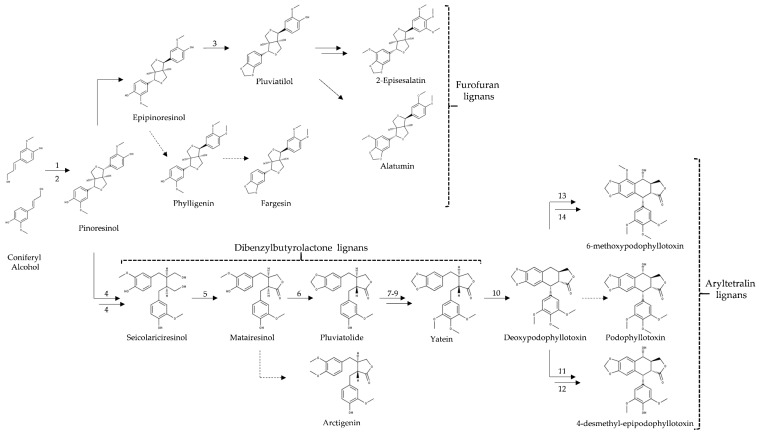
Biosynthetic pathway of selected lignans in different plant species. 1 = laccase, 2 = dirigent protein, 3 = CYP81Q3 in *S. alatum* [89], 4 = PLR1 in *L. perenne* [243], 5 = SDH in *F. intermedia* [244], 6 = CYP719A23 in *P. hexandrum* [249], 7–12 = OMT3, CYP71Cu1, OMT1, 2-ODD, CYP71BE54 and CYP82D61 in *P. hexandrum*, 13–14 = DOP6H, βP6OMT in *L. flavum*. Identified and postulated conversions are represented by solid and broken lines, respectively. Two arrows indicate several consecutive steps.

**Table 1 molecules-26-00883-t001:** Aglycones of lignans found in *Sesamum indicum* and related species.

Lignan	MW	Monoisotopic Mass	UV Maxima	Extinction Coefficient [g^−1^ L]	Content in Seeds [mg/100 g]	Reference
Sesamin	354.35	354.1103	287/236	23.03/26.01 ^4^	77–930	[46,47,48,49,50,51,52,53,54,55,56,57,58,59,60,61,62,63,64,65,66,67]
Sesamolin	370.35	370.1053	288/235	21.79/24.85 ^4^	61–530	[53,54,55,56,57,58,59,60,61,62,63,64,65,66,67,68,69,70,71,72]
Sesaminol	370.40	370.1053	238/295	3.99/3.85 ^5^	0–1.4	[73,74]
Sesamolinol	372.40	372.1209	231/287	3.95/3.80 ^5^	6–29	[53,75]
Pinoresinol	358.38	358.1416	232/280	-	29–38	[52,76,77,78]
Sesamol ^1^	138.12	138.0316	296/233	29.74/21.18 ^4^	0–30 ^1^	[62,63,79,80,81]
Lariciresinol	360.40	360.1572	230/280	-	0.8 ^3^	[77,78,82]
Matairesinol	358.40	358.1416	230/282	-	0.4 ^3^	[77]
Piperitol	356.40	356.1259	-	-	1.3 ^3^	[78]
Episesamin ^2^	354.40	354.1103	-	-	9–270	[54,73]
Samin	250.25	250.0841	287/238	-	50 ^2^	[83,84]

^1^ Only found in roasted seeds or oil. ^2^ Only found in bleached oil. ^3^ Only a single value reported. ^4^ In isooctane, ^5^ in chloroform.

**Table 2 molecules-26-00883-t002:** Lignans detected in *Sesamum* species other than *S. indicum*. The first report of each lignan in each species is cited.

Source	(+)-Sesamin	(+)-Sesamolin	(+)-Sesan-Golin	(+)-ala-Tumin	(+)-2-epi-Sesalatin	(+)-7′-epi-Sesantalin
*S. alatum*	[87] ^1^	[87] ^2^	[88]	[89]	[90]	
*S. angolense*	[66]	[66]	[91]			
*S. angustifolium*	[66]	[66]	[87]			
*S. orientale* *var. malabaricum*	[66] ^3^	[66] ^3^				
*S. calycinum*	[66]	[66]				
*S. latifolium*	[66]	[87] ^4^				
*S. radiatum*	[66]	[72] ^4,5^	[92]			[92]
*S. schinzianum*	[72] ^6^	[72] ^6^				
*S. mulayanum*	[93] ^7^	[93] ^7^				
*S. laciniatum*	[93] ^8^	[93] ^8^				
*S. capense*		[66] ^9^				
*S. pedalioides*	[66] ^9^					

^1^ [87] reported 0.01% in oil; [94] reported 0.14 mg g^−1^ seed; acc. to [93], the ratio of sesamin to sesamolin was 1:3. ^2^ [87] reported 0.01% in oil; [94] reported 0.38 mg g^−1^ seed. ^3^ Acc. to [93], the ratio of sesamin to sesamolin was 4:1. ^4^ In her publication from 2010 [95], Kamal-Eldin noted that *S. latifolium* was mistakenly named as *S. radiatum* in her previous publications, especially in [87]. For further details, see Section 2.8. ^5^ Acc. to [72], the ratio of sesamin to sesamolin was 7:1. ^6^ The ratio of sesamin to sesamolin was 5:1. ^7^ The ratio of sesamin to sesamolin was 5:1; authors indicated that *S. mulayanum* was a synonym for *S. malabaricum.*
^8^ The ratio of sesamin to sesamolin was 1:3. ^9^ Only trace amounts were detected in a single herbarium sample by TLC.

**Table 3 molecules-26-00883-t003:** Major glycosylated lignans found in *Sesamum indicum* and related species.

Lignan	MW	Monoisotopic Mass	Content in Seeds [mg/100 g]	Reference
Sesaminol monoglucoside	532.5	532.1581	5–20	[53,98,99]
Sesaminol diglucoside	694.6	694.2109	8–18	[53,98,99,100]
Sesaminol triglucoside	830.7	830.2481	14–91	[53,99,100]
Sesamolinol diglucoside	696.6	696.2265	5–232	[100] ^1^
Pinoresinol diglucoside ^2^	682.7	682.2473	1.4–2.1	[76,98]
Pinoresinol triglucoside	844.8	844.3001	5.22	[98]

^1^ First description and structure elucidation. ^2^ A mixture of three isomers: pinoresinol 4′-*O*-β-d-glucopyranosyl (l→6)-β-d-glucopyranoside, 4′-*O*-β-d-glucopyranosyl(l→2)-β-d-glucopyranoside, and di-*O*-β-d-glucoside.

**Table 4 molecules-26-00883-t004:** Lignans of *Sesamum indicum* found in tissues different from seeds.

Tissue	Lignans (µg/g)		Reference
	Sesamin	Sesamolin	
Callus	63–460	270–950	[224,225,226]
Leaves	2.6	-	[232]
Leaves, stem, roots, and flower	detected	detected	[232] *
Roots	0–220	not detected	[231]
Hairy roots	0–75	not detected	[231]

* The presence of the metabolites was claimed but no analytical data are provided.

**Table 5 molecules-26-00883-t005:** Proteins involved in the biosynthesis of sesame lignans.

Protein Name/Accession	Type of Enzyme	Function in the Pathway	EC Number
XP_011080883 *	Dirigent protein	Impart stereoselectivity on the phenoxy radical-coupling reaction	-
CYP81Q1	Cytochrome P450	Piperitol/sesamin synthase. Formation of dual methylenedioxy bridge	1.14.19.74
CYP92B14	Cytochrome P450	Sesamolin/sesaminol synthase.Converts sesamin to sesamolin or sesaminol	1.14.19.-
UGT71A9	Glycosyltransferase	Catalyzes the glucosylation at the 2-hydroxyl group of sesaminol	2.4.1.-
UGT94D1/UGT94AA2	Glycosyltransferase	Catalyzes the β1→6 glucosylation toward the sugar moiety of sesaminol 2-*O*-glucoside	2.4.1.-
UGT94AG1	Glycosyltransferase	Catalyzes the β1→2-*O*-glucosylation of sesaminol mono glucoside and sesaminol 2-*O*-β-d-glucosyl-(β1→6)-*O*-β-d-glucoside	2.4.1.-
XP_011092597 (SinPLR2) *	Bifunctional pinoresinol-lariciresinol reductase	Conversion of pinoresinol into lariciresinol and lariciresinol into secoisolariciresinol	1.23.1.1
XP_011094269 *	Secoisolariciresinol dehydrogenase-like	Conversion of secoisolariciresinol into matairesinol	1.1.1.331

* Predicted based on sequence similarity.

**Table 6 molecules-26-00883-t006:** Genes involved in the synthesis of lignans in sesame.

NCBI Accession Number or Gene ID	Gene Name	GC Content in Exons (%)	*Nc* Value *	Number of Introns	Linkage Group **
LOC105164033	----	52.0	56.06	0	LG6
AB194714	CYP81Q1	51.2	57.39	1	LG15
LC199944	CYP92B14	43.1	52.34	1	LG2
AB194716	CYP81Q3	51.3	57.02	0	(*S. alatum*)
AB293960	UGT71A9	47.6	59.01	0	LG16
AB333799	UGT94D1	49.8	53.13	0	LG4
LC484014	UGT94AA2	53.6	53.44	0	LG7
LC484013	UGT94AG1	40.6	50.50	0	LG10
LOC105172736	SinPLR2	45.2	57.93	3	LG10
LOC105174016	----	46.6	56.82	1	LG11

* *Nc* (effective number of codons) is a measure of the deviation of codon usage from the equal use of all codons [272]. ** The linkage group No. refers to *S. indicum*.

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
