# Peer review of "Lignans of Sesame (Sesamum indicum L.): A Comprehensive Review"

_molecules, 2021, doi:10.3390/molecules26040883_

Round 1

Reviewer 1 Report

Manuscript title: Lignans of Sesame (Sesamum indicum L.): A Compre3 hensive Review.

This article describes the chemical properties of lignans and their transformations during seed and sesame oil processing, their analysis, purification, and synthesis, their occurrence in Seseamum indicum and related plants, their biosynthesis, genetics, and biological activities.

It is of interest for the scientific community, well write, and rich in content. In my opinion, it is suitable for publication after a revision. Please, find below my suggestions:

1) The keywords should better represent the content of the review

2) The abstract must be more concise;

3) Introduction: the description of sesame (lines 29-39) must be improved; at the end of this paragraph, please, insert the period covered, the keywords used for the research, the databases used for the research, inclusion criteria, exclusion criteria;

4) The quality of Figures (the chemical structures of compounds are too small) must be improved;

5) A Conclusion Section must be inserted.

Reviewer 2 Report

The manuscript Lignans of Sesame (Sesamum indicum L.): A Comprehensive Review fits the journal’s scope. The review is organized and very well and clear written. Although some sections have a little too much history, the review is very useful to the health sciences professionals. The literature is well presented and the authors approach a critical view of it. Before publication the authors should improve the manuscript by adding a diagram regarding the sesame’s lignans medicinal findings. Also, the authors should check the manuscript for minor language/style/formatting errors.

Minor English/typing/formatting errors (rows 141, 511)

143-145  - please correct the phrase

Round 2

Reviewer 1 Report

This work is suitable for publication in this revised version.